# Reusing Combinatorial Structure: Faster Iterative Projections over Submodular Base Polytopes

**Jai Moondra**
Georgia Institute of Technology
`jmoondra3@gatech.edu`

**Hassan Mortagy**
Georgia Institute of Technology
`hmortagy@gatech.edu`

**Swati Gupta**
Georgia Institute of Technology
`swatig@gatech.edu`

## Abstract

Optimization algorithms such as projected Newton's method, FISTA, mirror descent and its variants enjoy near-optimal regret bounds and convergence rates, but suffer from a computational bottleneck of computing "projections" in potentially each iteration (e.g., $O(T^{1/2})$ regret of online mirror descent) [1, 2, 3, 4]. On the other hand, conditional gradient variants solve a linear optimization in each iteration, but result in suboptimal rates (e.g., $O(T^{2/3})$ regret of online Frank-Wolfe) [5, 6, 7]. Motivated by this trade-off in runtime v/s convergence rates, we consider iterative projections of close-by points over widely-prevalent submodular base polytopes $B(f)$. We develop a toolkit to speed up the computation of projections using both discrete and continuous perspectives (e.g., [8, 9, 10]). We subsequently adapt the away-step Frank-Wolfe algorithm to use this information and enable early termination. For the special case of cardinality based submodular polytopes, we improve the runtime of computing certain Bregman projections by a factor of $\Omega(n/\log(n))$. Our theoretical results show orders of magnitude reduction in runtime in preliminary computational experiments.

## 1  Introduction

Though the theory of discrete and continuous optimization methods has evolved independently over the last many years, machine learning applications have often brought the two regimes together to solve structured problems such as combinatorial online learning over rankings and permutations [11, 12, 13, 14], shortest-paths [15] and trees [16, 17], regularized structured regression [5], MAP inference, document summarization [18] (and references therein). One of the most prevalent forms of constrained optimization in machine learning is the use of iterative optimization methods such as online stochastic gradient descent, mirror descent variants, projected Newton's method, conditional gradient descent variants, fast iterative shrinkage-thresholding algorithm (FISTA). These methods repeatedly compute two main subproblems: either a projection (i.e., a convex minimization) or a linear optimization in each iteration. The former class of algorithms is known as projection-based optimization methods (e.g., projected Newton's method, see Table 1), and they enjoy near-optimal regret bounds in online optimization and near-optimal convergence rates in convex optimization compared to projection-free methods. These projection-based methods however suffer form high computational complexity per iteration due to the projection subproblem [1, 2, 19, 20, 4, 21]. E.g., online mirror descent is near-optimal in terms of regret (i.e., $O(\sqrt{T})$) for most online learning problems, however it is computationally restrictive for large scale problems [3]. On the other hand, online Frank-Wolfe is computationally efficient, but has a suboptimal regret of $O(T^{2/3})$ [7].

35th Conference on Neural Information Processing Systems (NeurIPS 2021).

| Algorithm | Subproblem solved | Steps for $\epsilon$-error |
|---|---|---|
| Vanilla Frank-Wolfe [5] | LO over polytope | $O\left(\frac{LD^2}{\epsilon}\right)$ |
| Away-steps Frank-Wolfe [6] | LO over polytope and active sets | $O\left(\kappa\left(\frac{D}{\delta}\right)^2 \log\frac{1}{\epsilon}\right)$ |
| *Projected gradient descent [24] | Euclidean projection over polytope | $O\left(\kappa \log\frac{1}{\epsilon}\right)$ |
| *Mirror descent (MD) [25] | Bregman Projection | $O\left(\kappa\nu^2 \log\frac{1}{\epsilon}\right)$ |
| *Projected Newton's method [24] | Euclidean projection over polytope scaled by (approximate) Hessian | $O\left((\kappa\beta)^3 \log\frac{1}{\epsilon}\right)$ |
| *Accelerated Proximal Gradient [26] | Euclidean projection over polytope | $O\left(\sqrt{\kappa} \log\frac{1}{\epsilon}\right)$ |
| *Fast Iterative Shrinkage-Thresholding Algorithm (FISTA) [27] | Euclidean projection over polytope | $O\left(\sqrt{\kappa} \log\frac{1}{\epsilon}\right)$ |

Table 1: Some iterative optimization algorithms which solve a linear or convex optimization problem in each iteration. Here, $\kappa := L/\mu$ is the condition number of the main optimization, $\nu$ is condition number of the mirror map used in MD, $D$ is the diameter of the domain, $\delta$ is the pyramidal width, $\beta \geq 1$ measures on how well the Hessian is approximated. Starred algorithms have dimension independent optimal convergence rates.

Discrete optimizers, in parallel, have developed beautiful characterizations of properties of convex minimizers over combinatorial polytopes, which typically results in non-iterative exact algorithms (upto solution of a univariate equation) for such polytopes. This theory however has not been properly integrated within the iterative optimization framework. Each subproblem within the above-mentioned iterative methods is typically solved from scratch, using a black-box subroutine, leaving a significant opportunity to speed-up "perturbed" subproblems using combinatorial structure. Motivated by these trade-offs in convergence guarantees and computational complexity, we ask if:

*Is it possible to speed up iterative subproblems of computing projections over combinatorial polytopes by reusing structural information from previous minimizers?*

This question becomes important in settings where the rate of convergence is more impactful than the time for computation, for e.g., regret impacts revenue for online retail platforms. However, the computational cost of solving a non-trivial projection sub-problem from scratch every iteration is the reason why these methods have remained of "theoretical" nature. We investigate if one can speed up iterative projections by reusing combinatorial information from past projections. Our techniques apply to iterative online and offline optimization methods such as Projected Newton's Method, Accelerated Proximal Gradient, FISTA, and mirror descent variants.

To give an example setup of our iterative framework, we consider the overarching optimization problem of minimizing a convex function $h : \mathcal{P} \to \mathbb{R}^n$ over a constrained set $\mathcal{P} \subseteq \mathbb{R}^n$ be $(P1)$, which we wish to solve using a regularized optimization method such as mirror descent and its variants. Typically, in such methods, iterates $x_t$ are obtained by taking an unconstrained gradient step, followed by a projection onto $\mathcal{P}$. We will refer to a subproblem of computing a single projection as (P2). Note that (P1) can be replaced by an online optimization problem as well, and similarly the iterative method to solve (P1) can be any one of those in Table 1.

$$(P1) \quad \begin{array}{l} \min h(x) \\ \text{subject to } x \in \mathcal{P} \end{array} \left.\right\} \begin{array}{l} \text{(P1) can be solved iteratively} \\ \text{using, e.g., mirror descent:} \end{array} \quad \begin{array}{l} 1. \ y_t = x_t - \gamma_t \nabla h(x_{t-1}) \\ 2. \ x_t = \arg\min_{z \in \mathcal{P}} D_\phi(z, y_t) \quad \textbf{(P2)} \end{array}$$

To solve (P2), we will typically aim for convex and discrete methods that can obtain arbitrary accuracy, to be able to bound errors in (P1). We will refer to iterates in (P1) as $x_1, x_2, \ldots x_t$, and if (P2) is solved using an iterative method like Away-step Frank-Wolfe [22, 23], we will refer to those iterates as $z^{(1)}, \ldots, z^{(k)}$ (depicted in Figure 1 (left, middle)). Our goal is to speed up the computation of $x_t$ by using the combinatorial structure of $x_1, \ldots, x_{t-1}, z^{(1)}, \ldots, z^{(k)}, y_1, \ldots, y_t$. To the best of our knowledge, we are the first to consider using the structure of previously projected points.

To capture a broad class of interesting combinatorial polytopes, we focus on submodular base polytopes. Submodularity is a discrete analogue of convexity, and captures the notion of diminishing returns. Submodular polytopes have been used in a wide variety of online and machine learning applications (see Table 2 in appendix). A typical example is when $B(f)$ is permutahedron, a polytope whose vertices are the permutations of $\{1, \ldots, n\}$, and is used for learning over rankings. Other machine learning applications include learning over spanning trees to reduce communication delays in networks, [12]), permutations to model scheduling delays [13], and $k$-sets for principal

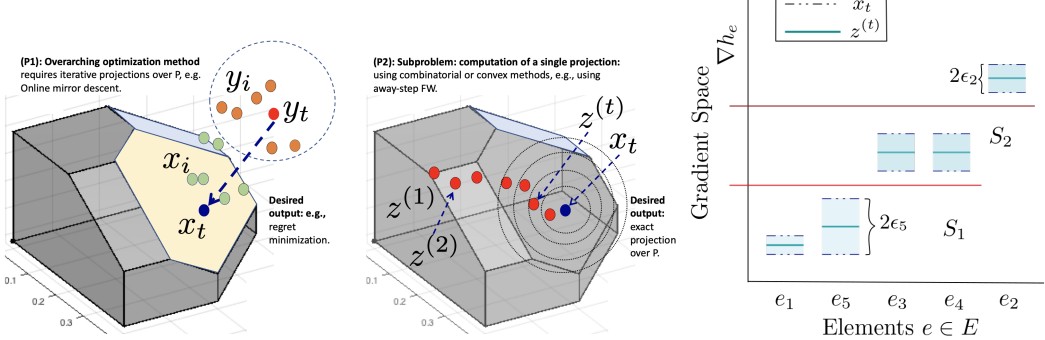

Figure 1: Left: **(P1)** represents an iterative optimization algorithm that computes projections $x_i$ for points $y_i$ in every iteration (see Table 1). Middle: **(P2)** represents subproblem of computing a single projection of $y_t$ using an iterative method with easier subproblems, e.g., away-step Frank-Wolfe where $z^{(i)}$ are iterates during a single run of AFW and converge to projection $x_t$ (of $y_t$). The goal is speed up the subproblems using both past projections $x_1, \ldots, x_{t-1}$, as well as iterates $z^{(1)}, \ldots, z^{(k)}$. Right: We show how to detect tight sets $S_1$ and $S_2$ for close-by points by looking at the maximum error in $\nabla h(x_t)$ (tools INFER1, INFER2).

component analysis [28], background subtraction in video processing and topographic dictionary learning [29], and structured sparse PCA [30]. Other example applications of convex minimization over submodular polytopes include computation of densest subgraphs [31], bounds on the partition function of log-submodular distributions [32] and distributed routing [33].

Though (Bregman) projections can be computed efficiently in closed form for certain simple polytopes (such as the $n$-dimensional simplex), the submodular base polytopes pose a unique challenge since they are defined using $2^n$ linear inequalities [34], and there exist instances with exponential extension complexity as well [35] (i.e., there exists no extended formulation with polynomial number of constraints for some submodular polytopes). Existing combinatorial algorithms for minimizing separable convex functions over base polytopes typically require iterative submodular function minimizations (SFM) [9, 8, 14], which are quite expensive in practice [36, 37]. However, these combinatorial methods highlight important structure in convex minimizers which can be exploited to speed up the continuous optimization methods.

In this paper, we bridge discrete and continuous optimization insights to speed up projections over submodular polytopes as follows:

(i) *Bregman Projections over cardinality-based polytopes*: We first show that the results of Lim and Wright [38] extend to all cardinality-based submodular polytopes (where $f(S) = g(|S|)$ for some concave function $g$) to give an $O(n \log n)$-time algorithm for computing a Bregman projection, improving the current best-known $O(n \log n + n^2)$ algorithm [14], in Section 3. These are exact algorithms (up to the solution of a univariate equation), compared to iterative continuous optimization methods.

(ii) *Toolkit for Exploiting Combinatorial Structure:* We next develop a toolkit (tools **T1-T6**) of provable ways for detecting tight inequalities, reusing active sets, restrict to optimal inequalities and rounding approximate projections to enable early termination:

   (a) INFER: We first show that for "close" points $y, \tilde{y}$ where the projection $\tilde{x}$ of $\tilde{y}$ on $B(f)$ is known, we can infer some tight sets for $x$ using the structure of $\tilde{x}$ without explicitly computing $x$ (**T1**). Further, suppose that we use a convergent iterative optimization method to solve the projection subproblem (P2) for $y_t$ to compute $x_t$, then given any iterate $z^{(k)}$ in such a method, we know that $\|z^{(k)} - x_t\| \leq \epsilon_k$ is bounded for strongly convex functions. Using this, we show how to infer some tight sets (provably) for $x_t$ for small enough $\epsilon_k$ (**T2**), in Section 4.1.

   (b) REUSE: Suppose we compute the projection $\tilde{x}$ of $\tilde{y}$ on $B(f)$ using AFW, and obtain an active set of vertices $A$ for $\tilde{x}$. Our next tool (**T3**) gives conditions under which $A$ is also an active set for $x$. Thus, $x$ can be computed by projecting $y$ onto $\text{Conv}(A)$ instead of $B(f)$ in Section 4.2.

   (c) RESTRICT: While solving the subproblem (P2), we show that discovered tight inequalities for the optimum solution can be incorporated into the linear optimization (LO) oracle over submodular polytopes, in Section 4.2. We modify Edmonds' greedy algorithm to do LO over any lower dimensional face of the submodular base polytope, while maintaining its efficient $O(n \log n)$ running time. Note that in general, while there may exist efficient algorithms to do

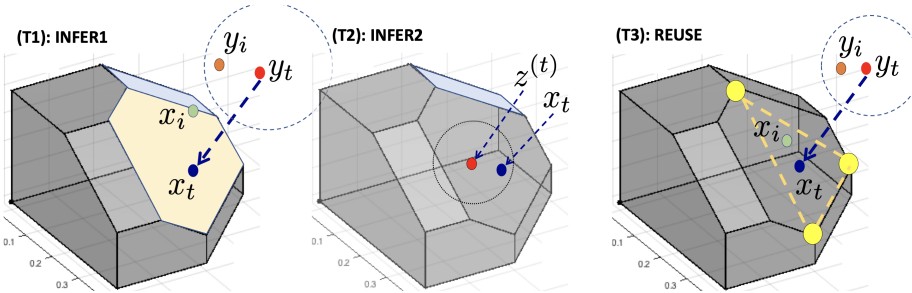

Figure 2: Toolkit to Speed Up Projections: INFER1 **(T1)** uses previously projected points to infer tight sets defining the optimal face of $x_t$ and is formally described by Theorem 3 (see also Figure 1-Right). On the other hand, INFER2 **(T2)** uses the closeness of iterates $z^{(t)}$ of an algorithm solving the projection subproblems (e.g. AFW) to the optimal $x_t$, to find more tight sets at $x_t$ (than those found by **(T1)**) (Lemma 4). REUSE **(T3)** uses active sets of previous projections computed using AFW (Lemma 1).

 LO over the entire polytope (e.g. shortest-paths polytope), restricting to lower dimensional faces may not be trivial.

(d) RELAX and ROUND: We give two approaches for rounding an approximate projection to an exact one in Section 4.3, which helps terminate iterative algorithms early. The first method uses INFER to iteratively finds tight sets at projection $x_t$, and then checks if we have found all such tight sets defining the optimal face by projecting onto the affine space of tight inequalities. If the affine projection $x_0$ is feasible in the base polytope, then this is optimal projection. The second rounding tool is algebraic in nature, and applicable only to base polytopes of integral submodular functions. It only requires a guarantee that the approximate projection be within a (Euclidean) distance of $1/(2n^2)$ to the optimal for Euclidean projections.

(iii) *Adaptive Away-Step Frank-Wolfe ($A^2$FW):* We combine the above-mentioned tools to give a novel adaptive away-step Frank-Wolfe variant in Section 5. We first use INFER **(T1)** to detect tight inequalities using past projections of $x_{t-1}$. Next, we start away-step FW to compute projection $x_t$ in iteration $t$ by REUSING the optimal active set from computation of $x_{t-1}$. During the course of $A^2$FW, we INFER tight inequalities iteratively using distance of iterates $z^{(t)}$ from optimal **(T2)**. To adapt to discovered tight inequalities, we use the modified greedy oracle **(T4)**. We check in each iteration if RELAX allows us to terminate early **(T5)**. In case of Euclidean projections, we also detect if rounding to lattice of feasible points is possible **(T6)**. We finally show an order of magnitude reduction in running time of online mirror descent by using $A^2$FW as a subroutine for computing projections in Section 5.1 and conclude with limitations in Section 5.2.

Although we show that our toolkit can help speed up iterative continuous optimization algorithms like mirror descent, the tools are more general and can be used to speed up other combinatorial algorithms like Groenvelt's Decomposition algorithm, Fujishige's minimum norm point, and Gupta et. al's Inc-Fix [39, 9, 14]. A special case of our rounding approach is used within the Fujishige-Wolfe minimum norm point algorithm to find approximate submodular function minimizers [40, 41].

Minimizing separable convex functions over submodular base polytopes was first studied by Fujishige [10] in 1980, followed by a series of results by Groenevelt [9], Hochbaum [42], and recently by Nagano and Aihara [8], and Gupta et. al. [43]. Each of these approaches considers different problem classes, but uses $O(n)$ calls to either parametric submodular function or submodular function minimization, with each computation discovering a tight set and reducing the subproblem size for future iterations. Both subroutines, however, can be expensive in practice. Frank-Wolfe variants on the other hand have attempted at incorporating geometry of the problem in various ways: restricting FW vertices to norm balls [44, 45, 46], or restricting away vertices to best possible active sets [47], or prioritizing in-face steps [48], or theoretical results such as [23] and [49] show that FW variants must use active sets that containing the optimal solution after crossing a polytope dependent radius of convergence. These results, however, do not use combinatorial properties of previous minimizers or detect tight sets with provable guarantees and round to those. To the best of our knowledge, we are the first to adapt away-step Frank-Wolfe to consider combinatorial structure from previous projections, and accordingly obtain improvements over the basic AFW algorithm. Although our $A^2$FW algorithm is most effective for computing projections (since we can invoke *all* our toolkit for projections, i.e.**(T1-T6)**), it is a standalone algorithm for convex optimization over base polytopes that enables early termination with the exact optimal solution (compared to the basic AFW) via rounding **(T5)** and improved convergence rates visa restricting **(T4)**. This might be of independent interest given the various applications mentioned above.

## 2  Preliminaries

Consider a compact and convex set $\mathcal{X} \subseteq \mathbb{R}^n$, and let $\mathcal{D} \subseteq \mathbb{R}^n$ be a convex set such that $\mathcal{X}$ is included in its closure. A mirror map $\phi : \mathcal{D} \to \mathbb{R}$ is a strictly (or $\mu$- strongly) convex[*] and continuously differentiable function over $\mathcal{D}$, and satisfies additional properties of divergence of the gradient on the boundary of $\mathcal{D}$, i.e., $\lim_{x \to \partial \mathcal{D}} \|\nabla \phi(x)\| = \infty$ (see [1, 20] for more details). We further assume that the mirror map $\phi$ is *uniformly* separable: $\phi = \sum_e \phi_e$ where $\phi_e : \mathcal{D}_e \to \mathbb{R}$ is the same function for all $e \in E$. We use $\|\cdot\|$ to denote the Euclidean norm unless otherwise stated. We say $\phi$ is $L$- smooth if $\|\nabla \phi(x) - \nabla \phi(z)\| \le L \|x - z\|$ for all $x, z \in \mathcal{D}$. The Bregman divergence generated by a mirror map $\phi$ is defined as $D_\phi(x, y) := \phi(x) - \phi(y) - \langle \nabla \phi(y), x - y \rangle$. For example, the Euclidean mirror map is given by $\phi = \frac{1}{2}\|x\|^2$, for $\mathcal{D} = \mathbb{R}^E$ and is 1-strongly convex with respect to the $\ell_2$ norm. In this case $D_\phi(x, y) = \frac{1}{2}\|x - y\|_2^2$ reduces to the Euclidean squared distance (see Table 3). We denote the Fenchel-conjugate of the divergence by $D_\phi^*(z, y) = \sup_{x \in \mathcal{D}}\{\langle z, x \rangle - D_\phi(x, y)\}$ for any $z \in \mathcal{D}^*$, where $\mathcal{D}^*$ is the dual space to $\mathcal{D}$ (in our case since $\mathcal{D} \subseteq \mathbb{R}^n$, $\mathcal{D}^*$ can also be identified with $\mathbb{R}^n$).

### Submodularity and Convex Minimizers over Base Polytopes

Let $f : 2^E \to \mathbb{R}$ be a submodular function defined on a ground set of elements $E$ ($|E| = n$), i.e. $f(A) + f(B) \ge f(A \cup B) + f(A \cap B)$ for all $A, B \subseteq E$. Assume without loss of generality that $f(\emptyset) = 0$, $f(A) > 0$ for $A \ne \emptyset$ and that $f$ is monotone[†]. We denote by $EO$ the time taken to evaluate $f$ on any set. For $x \in \mathbb{R}^E$, we use the shorthand $x(S)$ for $\sum_{e \in S} x(e)$, and by both $x(e)$ and $x_e$ we mean the value of $x$ on element $e$. Given such a submodular function $f$, the polymatroid is defined as $P(f) = \{x \in \mathbb{R}_+^E : x(S) \le f(S) \forall S \subseteq E\}$ and the base polytope as $B(f) = \{x \in \mathbb{R}_+^E : x(S) \le f(S) \forall S \subset E, \ x(E) = f(E)\}$ [51]. A typical example is when $f$ is the rank function of a matroid, and the corresponding base polytope corresponds to the convex hull of its bases (see Table 2).

Consider a submodular function $f : 2^E \to \mathbb{R}$ with $f(\emptyset) = 0$, and let $c \in \mathbb{R}^n$. Edmonds gave the greedy algorithm to perform linear optimization $\max c^T x$ over submodular base polytopes for monotone submodular functions. Order elements in $E = \{e_1, \ldots, e_n\}$ such that $c(e_i) \ge c(e_j)$ for all $i < j$. Define $U_i = \{e_1, \ldots, e_i\}$, and let $x^*(e_j) = f(U_j) - f(U_{j-1})$. Then, $x^* = \max_{x \in B(f)} c^\top x$. Further, we will use the following characterization of convex minimizers over base polytopes:

**Theorem 1** (Theorem 4 in [14]). *Consider any continuously differentiable and strictly convex function $h : \mathcal{D} \to \mathbb{R}$ and submodular function $f : 2^E \to \mathbb{R}$ with $f(\emptyset) = 0$. Assume that $B(f) \cap \mathcal{D} \ne \emptyset$. For any $x^* \in \mathbb{R}^E$, let $F_1, F_2, \ldots, F_l$ be a partition of the ground set $E$ such that $(\nabla h(x^*))_e = c_i$ for all $e \in F_i$ and $c_i < c_l$ for $i < l$. Then $x^* = \arg\min_{x \in B(f)} h(x)$ if and only if $x^*$ lies on the face $H^*$ of $B(f)$ given by $H^* := \{x \in B(f) \mid x(F_1 \cup F_2 \cup \cdots \cup F_i) = f(F_1 \cup F_2 \cup \cdots \cup F_i) \forall 1 \le i \le l\}$.*

To see why this holds, note that the first-order optimal condition for convex optimization gives us the following certificate $x^* = \arg\min_{x \in B(f)} h(x) \Leftrightarrow \nabla h(x^*)^T(z - x^* \ge 0 \ \forall z \in B(f) \Leftrightarrow x^* \in \arg\min_{z \in B(f)} \nabla h(x^*)^T z$. The theorem then follows by applying Edmond's greedy algorithm to $\arg\min_{z \in B(f)} \nabla h(x^*)^T z$ to obtain the levels of the partial derivatives of $x^*$ as $F_1, F_2, \ldots F_k$, which form the optimal face $H^*$ of $x^*$. For separable convex functions like Bregman divergences (in Table 3), we can thus compute $x^*$ by solving univariate equations in a single variable if the tight sets $F_1, \ldots, F_k$ of $x^*$ are known. We equivalently refer to corresponding inequalities $x(F_i) = f(F_i)$ as the optimal tight inequalities.

## 3  Bregman Projections over Cardinality-based Submodular Polytopes

We first improve the runtime of exact combinatorial algorithms for computing uniform Bregman projections over cardinality-based submodular polytopes. The key observation that allows us to do that is the following generalization of Lim and Wright's result [38], which, to the best of our knowledge is the first result to explicitly state the relation between Bregman projections on general cardinality-based submodular polytopes and isotonic optimization:

---

[*]A differentiable function $h$ is said to be strictly convex over domain $\mathcal{D}$ if $h(y) > h(x) + \langle \nabla h(x), y - x \rangle$ for all $x, y \in \mathcal{D}$. Moreover, a differentiable function $h$ is said to be $\mu$-strongly convex over domain $\mathcal{D}$ with respect to a norm $\|\cdot\|$ if $h(y) \ge h(x) + \langle \nabla h(x), y - x \rangle + \frac{\mu}{2}\|y - x\|^2$ for all $x, y \in \mathcal{D}$.

[†]$f$ is monotone if $f(A) \le f(B) \ \forall A \subseteq B \subseteq E$. For any non-negative submodular function $f$, we can consider a corresponding monotone submodular function $\bar{f}$ such that $P(f) = P(\bar{f})$ (see Section 44.4 of [50]).

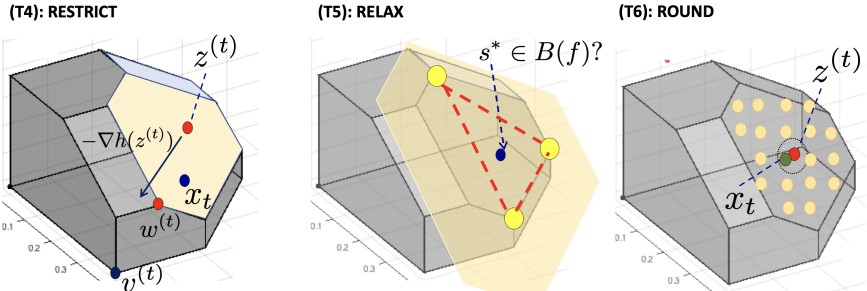

Figure 3: Proposed Toolkit (contd): RESTRICT **(T4)** restricts the LO oracle in AFW to the lower dimensional face defined by the tight sets found by **(T1)** and **(T2)** (Theorems 5, 6). Note that the restricted vertex $w^{(t)}$ gives better progress than the orginal FW vertex $v^{(t)}$. RELAX **(T5)** enables early termination of algorithms solving projection subproblems (e.g. AFW) as soon as all tight sets defining the optimal face are found (Theorem 2). Finally, ROUND **(T6)** gives an integral rounding approach for special cases (Lemma 3).

**Theorem 2** (Dual of projection is isotonic optimization). *Let $f : 2^E \to \mathbb{R}$ be a cardinality-based monotone submodular function, that is $f(S) = g(|S|)$ function for some nondecreasing concave function g. Let $c_i := g(i) - g(i-1)$ for all $i \in [E]$. Let $\phi : \mathcal{D} \to \mathbb{R}$ be a strictly convex and uniformly seperable mirror map. Let $B(f) \cap \mathcal{D} \neq \emptyset$ and consider any $y \in \mathbb{R}^n$. Let $\{e_1, \ldots, e_n\}$ be an ordering of the ground set E such that $y_1 \geq \cdots \geq y_n$. Then, the following problems are primal-dual pairs*

$$(P) \quad \begin{array}{l} \min \ D_\phi(x,y) \\ \text{subject to } x \in B(f) \end{array} \qquad (D) \quad \begin{array}{l} \max \ -D_\phi^*(z,y) + z^T c \\ \text{subject to } z_1 \leq \cdots \leq z_n \end{array}. \quad (1)$$

*Moreover, from a dual optimal solution $z^*$, we can recover the optimal primal solution $x^*$.*

To prove this result, we derive the Fenchel dual problem $(D)$ by using the structure of cardinality-based polytopes, and restricting the minimizer to the optimal face (see Appendix C). Problem $(D)$ in (1) is in fact a separable isotonic optimization problem[‡], which highlights an interesting connection between projections on cardinality-based polytopes [52, 53, 18]. In particular, when $\phi(x) = \frac{1}{2}\|x\|^2$, the dual problem $(D)$ in (1) becomes the following $\min_z \{\frac{1}{2}\|z - (c - y)\|^2 \mid z_1 \leq \cdots \leq z_n\}$ isotonic regression problem. Learning over projections is therefore dual to performing isotonic regression for perturbed data sets. Using the same algorithm as Lim and Wright's, i.e., the Pool Adjacent Violators (PAV) [54], we can solve the dual problem $(D)$ with a faster running time of $O(n \log n + nEO)$ compared to $O(n^2 + nEO)$ of [43]. We include the details about the algorithm and correctness in Appendix C. It is worth noting that linear optimization over $B(f)$ also has a running time of $O(n \log n + nEO)$ using Edmonds' greedy algorithm [34]. Therefore, for cardinality-based polytopes, when solving the projection sub-problem (P2), it is better to use a combinatorial algorithm (e.g. PAV) than any iterative optimization method (e.g. FW). Note that any FW iteration needs to sort the gradient vector (i.e., linear optimization over the base polytope) which is also $O(n \log n)$ in runtime. For cardinality-based polytopes, therefore, projection-based methods to solve (P1) are computationally competitive with conditional gradient methods.

## 4 Toolkit to Adapt to Previous Combinatorial Structure

In the previous section, we gave an $O(n \log n)$ exact algorithm for computing Bregman projections over cardinality-based polytopes. However, the pool-adjacent-violator algorithm is very specific to the cardinality-based polytopes and does not extend to general submodular polyhedra. To compute a projection over the challenging submodular base polytope, there are currently only two potential ways of doing so: (i) using Frank-Wolfe variants (due to simple linear sub-problems), (ii) using combinatorial algorithms such as those of [9, 8] (which typically rely on submodular function minimization for detecting tight sets). In this section, we construct a toolkit to speed up these approaches, and consequently speed up iterative projections over general submodular polytopes.

### 4.1 INFER **tight inequalities**

We first present our INFER tool **T1** that recovers some tight inequalities of projection of $\tilde{y}$ by using the tight inequalities of the projection of a close-by perturbed point $y \in \mathbb{R}^n$. The motivation of this result stems from the fact that projection-based optimization methods often move slowly, i.e., points

---

[‡]A separable isotonic optimization problem is of the form $\min \sum_{i=1}^n h_i(x_i)$ subject to $x_1 \leq x_2 \leq \cdots \leq x_n$, where $h_i$ are univariate strictly convex functions

$y, \tilde{y}$ to be projected are often close to each other, and so are their corresponding projections $x, \tilde{x}$. Our first result is specifically for Euclidean projections.

**Theorem 3** (Recovering tight sets from previous projections **(T1)**). *Let $f : 2^E \to \mathbb{R}$ be a monotone submodular function with $f(\emptyset) = 0$. Further, let $y$ and $\tilde{y} \in \mathbb{R}^E$ be such that $\|y - \tilde{y}\| \le \epsilon$, and $x, \tilde{x}$ be the Euclidean projections of $y, \tilde{y}$ on $B(f)$ respectively. Let $F_1, F_2, \ldots, F_k$ be a partition of the ground set $E$ such that $x_e - y_e = c_i$ for all $e \in F_i$ and $c_i < c_l$ for $i < l$. If $c_{j+1} - c_j > 4\epsilon$ for some $j \in [k-1]$, then the set $S = F_1 \cup \cdots \cup F_j$ is also a tight set for $\tilde{x}$, i.e. $\tilde{x}(S) = f(S)$.*

Note that $x_e - y_e$ is the partial derivative of the distance function from $y$ at $x$. The proof shows that for $e \in E$, $\tilde{x}_e - \tilde{y}_e$ is close to $x_e - y_e$ and relies on the smoothness and non-expansivity of Euclidean projection. This helps us infer that the relative order of coordinates in $\tilde{x} - \tilde{y}$ (i.e., the coordinate-wise partial derivatives) is close to the relative order of coordinates in $x - y$. This relative order then determines tight sets for $x$, due to first-order optimality characterization of Theorem 1. See Appendix D.2 for a complete proof, where we also generalize the theorem to any Bregman projection that is $L$-smooth and non-expansive. In Section 5.1, we will show that this theorem infers most of the tight inequalities computationally (see Figure 4-left).

Next, consider the subproblem (P2) of computing the projection $x_t$ of a point $y_t$. Let $z^{(k)}$ be the iterates in the subproblem that are convergent to $x_t$. The points $z^{(k)}$ grow progressively closer to $x_t$, and our next tool INFER **T2** helps us recover tight sets for $x_t$ using the gradients of points $z^{(k)}$.

**Theorem 4** (Adaptively inferring the optimal face **(T2)**). *Let $f : 2^E \to \mathbb{R}$ be monotone submodular with $f(\emptyset) = 0$, $h : \mathcal{D} \to \mathbb{R}$ be a strictly convex and $L$-smooth function, where $B(f) \cap \mathcal{D} \ne \emptyset$. Let $x := \arg\min_{z \in B(f)} h(z)$. Consider any $z \in B(f)$ such that $\|z - x\| \le \epsilon$. Let $\tilde{F}_1, \tilde{F}_2, \ldots, \tilde{F}_k$ be a partition of the ground set $E$ such that $(\nabla h(z))_e = \tilde{c}_i$ for all $e \in \tilde{F}_i$ and $\tilde{c}_i < \tilde{c}_l$ for $i < l$. Suppose $\tilde{c}_{j+1} - \tilde{c}_j > 2L\epsilon$ for some $j \in [k-1]$. Then, $S = F_1 \cup \cdots \cup F_j$ is tight for $x$, i.e. $x(S) = f(S)$.*

The proof of this theorem, similar to Theorem 3, relies on the $L$-smoothness of $h$ to show that the relative order of coordinates in $\nabla h(x_t)$ is close to the relative order of coordinates in $\nabla h(z^{(k)})$, which helps infer some tight sets for $x$. See Appendix D.2 for a complete proof and Figure 1-right for an example. Note that while Theorem 3 is restricted to Euclidean projections, Theorem 4 applies to any smooth strictly convex function.

## 4.2 ReUse and Restrict

We now consider computing a single projection (P2) using Frank-Wolfe variants, that have two main advantages: (i) they maintain an active set for their iterates as a (sparse) convex combination of vertices, (ii) they only solve LO every iteration. Our first REUSE tool gives conditions under which a new projection has the same active set $A$ as a point previously projected, which allows for a faster projection onto the convex hull of $A$ (proof is included in Appendix D.2).

**Lemma 1** (Reusing active sets **(T3)**). *Let $\mathcal{P} \subseteq \mathbb{R}^n$ be a polytope with vertex set $\text{vert}(\mathcal{P})$. Let $x$ be the Euclidean projection of some $y \in \mathbb{R}^n$ on $\mathcal{P}$. Let $\mathcal{A} = \{v_1, \ldots, v_k\} \subseteq \text{vert}(\mathcal{P})$ be an active set for $x$, i.e., $x = \sum_{i \in [k]} \lambda_i v_i$ for $\|\lambda\|_1 = 1$ and $\lambda > 0$. Let $F$ be the minimal face of $x$ and $\Delta := \min_{v \in \partial \text{Conv}(\mathcal{A})} \|x - v\|$ be the minimum distance between $x$ and the boundary of $\text{Conv}(\mathcal{A})$. Then, $\mathcal{A}$ is also an active set for the Euclidean projection of any point $\tilde{y} \in \mathbb{B}_\Delta(y) \cap \text{Cone}(F)$, where $\mathbb{B}_\Delta(y) = \{\tilde{y} \in \mathbb{R}^n \mid \|\tilde{y} - y\| \le \min\{\Delta, \|x - y\|\}\}$ is a closed ball centered at $y$.*

In the previous section, we presented combinatorial tools to detect tight sets at the optimal solution. We now use our RESTRICT tool to strengthen the LO oracle in FW by restricting it to the lower dimensional faces defined by the tight sets we found (instead of doing LO over the whole polytope). Note that doing linear optimization over lower dimensional faces of polytopes, in general, is significantly harder (e.g., for shortest paths polytope). For submodular polytopes however, we show that we can do LO over any face of $B(f)$ efficiently using a modified greedy algorithm (Algorithm 2 in Appendix B). Given a set of tight inequalities, one can uncross these to form a *chain* of tight sets, i.e., any face of $B(f)$ can be written using a chain of subsets that are tight (see e.g. Section 44.6 in [55]). Given such a chain, our modified greedy algorithm then orders the cost vector in decreasing order so that it respects a given tight chain family of subsets. Once it has that ordering, it proceeds in the same way as in Edmonds' greedy algorithm [34]. We include a proof of the following theorem in Appendix D.2.

**Theorem 5** (Linear optimization over faces of $B(f)$ **(T4)**). *Let $f : 2^E \to \mathbb{R}$ be a monotone submodular function with $f(\emptyset) = 0$. Further, let $F = \{x \in B(f) \mid x(S_i) = f(S_i) \text{ for } S_i \in \mathcal{S}\}$ be a*

*face of $B(f)$, where $\mathcal{S} = \{S_1, \ldots S_k | S_1 \subseteq S_2 \ldots \subseteq S_k\}$. Then the modified greedy algorithm (Alg. 2) returns $x^* = \arg\max_{x \in F} \langle c, x \rangle$ in $O(n \log n + nEO)$ time.*

### 4.3 Rounding

Approximation errors in projection subproblems often impact (adversely) the convergence rate of the overarching iterative method unless the errors decrease at a sufficient rate [56, 57]. Our goal in this section is to detect if all tight sets at the optimum have been inferred, and enable early termination by computing the exact minimizer. In 2020, [58] gave primal gap bounds after which away-step FW reaches the optimal face, assuming strict complementarity assumption which need not hold even for computing a Euclidean projection. Further, [59], showed that there exists some convergence radius $R$ such that for any iterate $z^{(t)}$ of AFW, if $\|z^{(t)} - x^*\| \le R$, then any active set for $z^{(t)}$ must contain $x^*$, but the parameter $R$ existential and is non-trivial to compute. We complement these results by rounding our approximate projections to an exact one based on structure in partial derivatives.

Suppose that we have a candidate chain $\mathcal{S} = \{S_1, \ldots S_k\}$ of tight sets (e.g., using INFER). We observe that if the affine minimizer over $\mathcal{S}$, i.e., $\tilde{x} := \arg\min\{h(x) \mid x(S) = f(S) \forall S \in \mathcal{S}\}$ is feasible in $B(f)$, then this is indeed the optimum solution $\tilde{x} = x^*$.

**Lemma 2** (Rounding to optimal face **(T5)**). *Let $f : 2^E \to \mathbb{R}$ be a monotone submodular function with $f(\emptyset) = 0$. Let $h : \mathcal{D} \to \mathbb{R}$ be a strictly convex, where $B(f) \cap \mathcal{D} \ne \emptyset$. Let $x^* := \arg\min_{x \in B(f)} h(x)$, and let $\mathcal{S} = \{S_1, \ldots S_k\}$ contain some of the tight sets at $x^*$, i.e. $x^*(S_i) = f(S_i)$ for all $i \in [k]$. Further, let $\tilde{x} := \arg\min\{h(x) \mid x(S) = f(S) \forall S \in \mathcal{S}\}$ be the optimal solution restricted to the face defined by the tight set inequalities corresponding to $\mathcal{S}$. Then, $x^* = \tilde{x}$ iff $\tilde{x}$ is feasible in $B(f)$. In particular, if $\mathcal{S}$ contains all the tight sets at $x^*$, then $x^* = \tilde{x}$.*

The proof of this lemma can be found in Appendix D.3, and as a subroutine in Appendix B. We note that this holds for *any polytope*: if we know that tight inequalities at the minimizer we can restrict the optimization problem to the face defined by those tight inequalities and ignore the other constraints defining the polytope (see Lemma 4 in Appendix C). To check whether $\tilde{x} \in B(f)$ in general requires an expensive submodular function minimization, but instead we just check whether $\tilde{x}$ is in the convex hull of $\{v^{(1)}, \ldots, v^{(t)}\}$, where $v^{(i)}$ are the FW vertices of $B(f)$ that we have computed in Line 3 of Algorithm A$^2$FW up to iteration $t$. Using [59], we know that there will be a point at which the optimal solution is contained in the current active set.

We now present our second rounding tool ROUND for base polytopes of integral submodular functions. It only requires a guarantee that the approximate projection be within a (Euclidean) distance of $1/(2|E|^2)$ to the optimal projection. This generalizes the robust version of Fujishige's theorem given in [41], connecting the MNP over $B(f)$ and the set minimizing the submodular function value.

**Lemma 3** (Combinatorial Integer Rounding Euclidean Projections **(T6)**). *Let $f : 2^E \to \mathbb{Z} \, (|E| = n)$ be a monotone submodular function with $f(\emptyset) = 0$. Consider $y \in \mathbb{Z}^E$ and let $h(x) = \frac{1}{2}\|x - y\|^2$. Let $x^* := \arg\min_{x \in B(f)} h(x)$. Consider any $x \in B(f)$ such that $\|x - x^*\| < \frac{1}{2n^2}$. Define $Q := \mathbb{Z} \cup \frac{1}{2}\mathbb{Z} \cup \ldots \cup \frac{1}{n}\mathbb{Z}$, and for any $r \in \mathbb{R}$, let $q(r) := \arg\min_{s \in Q} |r - s|$. Then, $q(x_e)$ is unique for all $e \in E$, and the optimal solution is given by $x_e^* = q(x_e)$ for all $e \in E$.*

This rounding algorithm runs in time $O(n^2 \log n)$ and is given in Algorithm 5 in Appendix B. The proof proceeds by showing that $x_e^* \in S$ for all $e \in E$, and that the distance between two points in $S$ is at least $\frac{1}{|E|^2}$, so that one can always round to $x^*$ correctly (complete proof is in Appendix D.3).

## 5  Adaptive Away-steps Frank-Wolfe (A$^2$FW)

We are now ready to present our Adaptive AFW (Alg. 1) by combining tools presented in the previous section. First using the INFER1, we detect some of the tight sets $\mathcal{S}$ at the optimal solution before even running A$^2$FW, and accordingly warm-start A$^2$FW with $z^{(0)}$ in the tight face of $\mathcal{S}$. A$^2$FW operates similar to the away-step Frank-Wolfe, but during the course of the algorithm it restricts to tight faces as it discovers them (using INFER2), adapts the linear optimization oracle (using RESTRICT), and attempts to round to optimum (using ROUND, RELAX). To apply INFER2 (subroutine included as Algorithm 3), consider an iteration $t$ of A$^2$FW, where we have computed the FW gap $g_t^{\text{FW}} := \max_{v \in B(f)} \langle -\nabla h(z^{(t)}), v - z^{(t)} \rangle$ (see line 11 in Algorithm 1). For $\mu$-strongly convex $h$, we have:

$$\frac{\mu}{2}\|z^{(t)} - x^*\|^2 \le h(z^{(t)}) - h(x^*) \le \max \langle -\nabla h(z^{(t)}), v - z^{(t)} \rangle = g_t^{\text{FW}}, \qquad (2)$$

---

**Algorithm 1** Adaptive Away-steps Frank-Wolfe ($A^2$FW)

---

**Input:** Submodular $f : 2^E \to \mathbb{R}$, $(\mu, L)$-strongly convex and smooth $h : B(f) \to \mathbb{R}$, chain of tight cuts $\mathcal{S}$
(e.g., using INFER1), $z^{(0)} \in B(f) \cap \{x(S) = f(S), S \in \mathcal{S}\}$ with active set $\mathcal{A}_0$, tolerance $\varepsilon$.

1: Initialize $t = 0, g_0^{\text{FW}} = +\infty, v^{(0)} = z^{(0)}$
2: **while** $g_t^{\text{FW}} \geq \varepsilon$ **do**
3:     $\mathcal{S}_{new} = \mathcal{S} \cup \text{INFER2}(h, z^{(t)}, 2L\sqrt{2g_t^{\text{FW}}/\mu})$          ▷ use toolkit to find new tight sets
4:     $\tilde{x}, Flag = \text{RELAX}(\mathcal{S}_{new}, \{v^{(0)} \dots v^{(t)}\})$
5:     **if** $Flag = True$, **return** $\tilde{x}$
6:     **if** $|\mathcal{S}_{new}| > |\mathcal{S}|$ **then**
7:        Set $z^{(t+1)} \in \arg\min_{v \in F(\mathcal{S}_{new})} \left\langle \nabla h(z^{(t)}), v \right\rangle$ and $\mathcal{A}_{t+1} = z^{(t+1)}$        ▷ round and restart
8:     **else**                                            ▷ do iteration of AFW restricted to $F(\mathcal{S})$
9:        Compute $v^{(t)} \in \arg\min_{v \in F(\mathcal{S})} \left\langle \nabla h(z^{(t)}), v \right\rangle$             ▷ use toolkit
10:       Compute away-vertex $a^{(t)} \in \arg\max_{v \in \mathcal{A}_t} \left\langle \nabla h(z^{(t)}), v \right\rangle$
11:       $z^{(t+1)}, \mathcal{A}_{t+1}, g_{t+1}^{\text{FW}} = AFW\text{-}update(z^{(t)}, v^{(t)}, a^{(t)}, \mathcal{A}_t)$
12:     **end if**
13:     Update $t := t + 1$ and $\mathcal{S} = \mathcal{S}_{new}$
14: **end while**
**Return:** $z^{(t)}$

---

and so $\|z^{(t)} - x^*\| \leq \sqrt{2g_t^{\text{FW}}/\mu}$. Let $\tilde{F}_1, \tilde{F}_2, \dots, \tilde{F}_k$ be a partition of the ground set $E$ such that $(\nabla h(z^{(t)}))_e = \tilde{c}_i$ for all $e \in F_i$ and $\tilde{c}_i < \tilde{c}_l$ for all $i < l$. If $\tilde{c}_{j+1} - \tilde{c}_j > 2L\sqrt{2g_t^{\text{FW}}/\mu}$ for some $j \in [k-1]$, then Theorem 4 implies that $S = F_1 \cup \dots \cup F_j$ is tight for $x^*$, i.e. $x^*(S) = f(S)$.

Overall in $A^2$FW, we maintain a set $\mathcal{S}$ containing all such tight sets $S$ at the optimal solution that we have found so far. We use those tight sets as follows: (i) we restrict our LO oracle to the lower dimensional face we identified using the modified greedy algorithm (RESTRICT- **(T4)**). (ii) We use our RELAX (**(T5)**) tool to check weather we have identified all the tight-sets defining the optimal face (Lemma 2). If yes, then we round the current iterate to the optimal face and terminate the algorithm early. For (Euclidean) projections over an integral submodular polytope, we can also use our ROUND **(T6)** tool to round an iterate close to optimal without knowing the tight sets. Whenever the algorithm detects a new chain of tight sets $\mathcal{S}_{new}$, it is restarted from a vertex in $F(\mathcal{S}_{new})$, which possibly has a higher function value than the current iterate. However, this increase in the primal gap is bounded as $h$ is finite over $B(f)$ and can happen at most $n$ times; thus, these restarts do not impact the convergence rate. The pseudocode of $A^2$FW is included in Algorithm 1.

**Convergence Rate:** As depicted in **(T4)** in Figure 3, restricting FW vertices to the optimal face results in better progress per iteration during the latter runs of the algorithm. The convergence rate of $A^2$FW depends on a geometric constant $\delta$ called the pyramidal width [6]. This constant is computed over the worst case face of the polytope. By iterative restricting the linear optimization oracle to optimal faces, we improve this worst case dependence in the convergence rate (proof in Appendix F):

**Theorem 6** (Convergence rate of $A^2$FW). *Let $f : 2^E \to \mathbb{R}$ be a monotone submodular function with $f(\emptyset) = 0$ and $f$ monotone. Consider any smooth strongly convex function $h(\cdot)$ with unique optimal $x^* \in B(f)$. Let $\mathcal{S}$ be the tight sets found up to iteration $t$ and $F(\mathcal{S})$ be the face defined by these tight sets. Then, the primal gap $w(z^{(t+1)}) := h(z^{(t+1)}) - h(x^*)$ of $A^2$FW decreases geometrically at each step that is not a drop step$^\S$ nor a restart step:*

$$w(z^{(t+1)}) \leq \left(1 - \frac{\mu\rho_{F(\mathcal{S})}^2}{4LD^2}\right) w(z^{(t)}), \text{where } D \text{ is the diameter of } B(f) \text{ and} \tag{3}$$

$\rho_{F(\mathcal{S})}$ *is the pyramidal width of $B(f)$ restricted to $F(\mathcal{S})$ (as defined by (24)). Moreover, in the worst case, the number of iterations to get an $\epsilon$-accurate solution is $O\left((nLD^2/(\mu\rho_{B(f)})^2)\log(1/\epsilon)\right)$.*

Note that $\rho_{F(\mathcal{S})}$ can be strictly larger than the worst-case pyramidal width over the entire polytope. For example, for the probability simplex (a submodular polytope; see Table 2), the pyramidal width restricted to a face $F$ is $2/\sqrt{\dim(F)}$ (assuming $\dim(F)$ is even for simplicity) [60]. To the best of our knowledge, we are the first to adapt AFW to tight faces as they are detected. This might be of independent interest to the SFM community.

---

$^\S$A drop step is when we take an away step with a maximal step size so that we drop a vertex from the current active set.

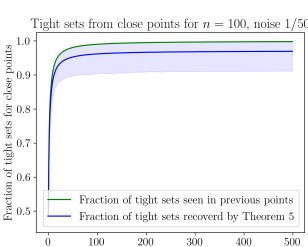 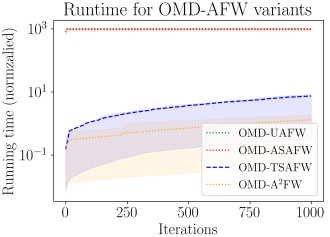 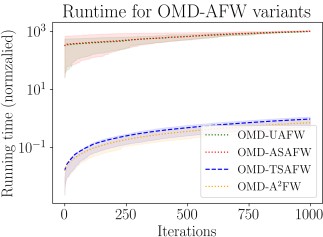

Figure 4: (left) 15-85% percentile plot of fraction of tight sets inferred by using INFER1 (blue) v/s highest number of tight sets common for $i$th iterate compared to previous $i-1$ iterates (in green) for close points generated randomly using Gaussian noise, over 500 runs. (middle) 25-75 percentile plots of normalized run times for OMD-AFW variants for first loss setting averaged over 20 runs. (right) 25-75 percentile plots of normalized run times for OMD-AFW variants with second loss setting averaged over 20 runs.

## 5.1 Computations

The code for our computations can be found on GitHub[¶]. In our first experiment, we iteratively compute the Euclidean projections of 500 randomly generated points on the permutahedron. The cloud of these 500 points is generated by fixing a random mean point and perturbing it using multivariate Gaussian noise with mean zero and standard deviation $\epsilon = 1/50$. We compute the projections of each point in the cloud exactly, and plot percentile plots of fraction of discovered tight sets from previous projections in Figure 4-left. The fraction of tight inequalities for each point $y_i$ that were already tight for some other previous point $y_0, \ldots, y_{i-1}$ is in green, the fraction of tight sets for $y_i$ inferred by using Theorem 3 is in blue. The plots average over 20 runs of this experiment. Note that our theoretical results give almost tight computational results, that is, we can recover most of the tight sets common between close points using Theorem 3.

In our second experiment (detailed in appendix G), motivated by the trade-off in regret versus time for online mirror descent and online Frank-Wolfe (OFW) variants, we conduct an experiment on the permutahedron $P$ with $n = 50$ elements. We consider a time horizon of $T = 1000$, and construct two noisy (linear) loss settings. For each of the two loss settings, we run Online Frank-Wolfe (OFW) and five variants of Online Mirror Descent (OMD) using the toolkit proposed: (1) OMD-UAFW: OMD with projection using vanilla away-step Frank-Wolfe (baseline), (2) OMD-ASAFW: OMD with AFW with reused active sets, (3) OMD-TSAFW: OMD with AFW with INFER, RESTRICT, and ROUNDING, (4) OMD-A²FW OMD with A²FW, and (5) OMD-PAV: OMD with PAV. We call the first four "OMD-AFW variants". Recall that OMD performs projections in potentially each iteration.

We normalized each OMD-UAFW run time to be 1000, and run times for all other variants in this run are correspondingly scaled in Figures 4-middle and 4-right. Each iteration of OMD involves projecting a point on the permutahedron, and the cumulative run times for these projections are plotted. The plots are averaged over 20 runs of this experiment for both the settings.

We see more than three orders of magnitude improvement in run time for OMD-ASAFW and OMD-A²FW compared to the unoptimized OMD-AFW. Both OMD-PAV and OFW run 4 to 6 orders of magnitudes faster on average than OMD-UAFW; however, OMD-PAV suffers from the limitation that it only applies to cardinality-based submodular polytopes, while OFW has significantly higher regret in computations. We summarize these results in Table 4 in Appendix G.

OMD has a regret 1 to 2 orders of magnitude lower than OFW on average, thus bolstering the claim that we need to invest research to speed-up this optimal learning method and its variants. This drop in regret is *significant* in terms of revenue for an online retail platform. The regret for all OMD variants was observed to be nearly the same. Overall, speeding up OMD is an example of the impact of our toolkit, which can be applied in the broader setting of iterative optimization methods.

## 5.2 Limitations and Open questions

There is still is a long way from closing the computational gap with Online Frank Wolfe. Our work inspires many future research questions, e.g., procedures to infer tight sets on non-submodular polytopes such as matchings and procedures to round iterates to the nearest tight face for combinatorial polytopes. We hope that our results can inspire future work that goes beyond looking at projection subroutines as black boxes. We believe that our work does not have any foreseeable negative ethical or societal impact.

---

[¶]`https://github.com/jaimoondra/submodular-polytope-projections`

## 6 Acknowledgments and Disclosure of Funding

The research presented in this paper was partially supported by the Georgia Institute of Technology ARC TRIAD fellowship and NSF grant CRII-1850182.

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
