base polytopes arising from submodular functions have been used to model combinatorial constraints in a wide variety of machine learning applications, such as MAP inference, document summarization, sensor placement, clustering, image segmentation [18]. In the table below we present some popular submodular functions and the problems arising from the corresponding submodular base polytopes:

| Problem | Submodular function, $S \subseteq E$ (unless specified) | Cardinality-based |
|---|---|---|
| $k$ out of $n$ experts ($k$-simplex), $E = [n]$ | $f(S) = \min\{|S|, k\}$ | ✓ |
| $k$-truncated permutations over $E = [n]$ | $f(S) = (n-k)|S|$ for $|S| \le k$, $f(S) = k(n-k) + \sum_{j=k+1}^{|S|}(n+1-s)$ if $|S| > k$ | ✓ |
| $k$-forests on $G = (V, E)$ | $f(S) = \min\{|V(S)| - \kappa(S), k\}$, $\kappa(S)$ is the number of connected components of $S$ | ✗ |
| Matroids over ground set $E$: $M = (E, \mathcal{I})$ | $f(S) = r_M(S)$, the rank function of $M$ | ✗ |
| Coverage of $T$: given $T_1, \ldots, T_n \subseteq T$ | $f(S) = |\cup_{i \in S} T_i|$, $E = \{1, \ldots, n\}$ | ✗ |
| Cut functions on a directed graph $D = (V, E)$, $c : E \to \mathbb{R}_+$ | $f(S) = c(\delta^{\text{out}}(S))$, $S \subseteq V$ | ✗ |

Table 2: Problems and the submodular functions (on ground set of elements $E$) that give rise to them.

Mirror descent variants compute a Bregman projection by minimizing Bregman divergence over $B(f)$. Bregman divergences are generated by a distance function or mirror map $\phi$ and the choice of the mirror map typically depends on the polytope given in the problem. In the table below, we present some popular uniform separable mirror maps and their corresponding divergences:

| Mirror Map $\phi(x) = \sum \phi_e(x_e)$ | $D_\phi(x, y)$ | Divergence |
|---|---|---|
| $\|x\|^2/2$ | $\sum_e (x_e - y_e)^2$ | Squared Euclidean Distance |
| $\sum_e x_e \log x_e - x_e$ | $\sum_e (x_e \log(x_e/y_e) - x_e + y_e)$ | Generalized KL-divergence |
| $-\sum_e \log x_e$ | $\sum_e (x_e \log(x_e/y_e) - x_e + y_e)$ | Itakura-Saito Distance |
| $\sum_e (x_e \log x_e + (1 - x_e) \log(1 - x_e))$ | $\sum_e (x_e \log(x_e/y_e) + (1 - x_e) \log((1 - x_e)/(1 - y_e))$ | Logistic Loss |

Table 3: Examples of some popular uniform separable mirror maps and their corresponding divergences.

# B    Algorithms

We first give our modified greedy algorithm for doing linear optimization over low dimensional faces of the base polytope. This tool is used as to compute FW vertices in lower dimensional faces within our A$^2$FW algorithm.

---

**Algorithm 2** Greedy algorithm for faces of $B(f)$

---

**Input:** Monotone submodular $f : 2^E \to \mathbb{R}$, objective $c \in \mathbb{R}^n$, face $F = \{x \in B(f) \mid x(S_i) = f(S_i)$, where $S_1 \subset \cdots \subset S_k = E$ where $S_i$ form a chain$\}$.
1: Consider an ordering on the ground set of elements $E = \{e_1, \ldots, e_n\}$ such that (i) it respects the given chain, i.e., $S_i = \{e_1, \ldots, e_{s_i}\}$ for all $i$, and (ii) each set $S_i \setminus S_{i-1} = \{e_{s_{i-1}+1}, \ldots, e_{s_i}\}$ is in decreasing order of cost, i.e., $c(e_{s_{i-1}+1}) \ge \ldots \ge c(e_{s_i})$.
2: Let $x^*(e) := f(\{e_1, \ldots, e_j\}) - f(\{e_1, \ldots, e_{j-1}\})$, for $i \in [n]$.
**Return:** $x^* = \arg\max_{x \in F} \langle c, x \rangle$

---

We next convert Theorem 4 and Lemmas 2, 3 to algorithm environments and include them in this section. First we present our INFER2 tool, which could be used to to detect tight sets at the optimal solution for any iterative algorithm used to compute a projection in problem (P2). For example, this tool is used as sub-routine in our A$^2$FW to find tight sets in AFW and make it adaptive.

---

**Algorithm 3** Detect Tight Sets (**T2**): INFER2$(h, z, \epsilon)$

---

**Input:** Submodular function $f : 2^E \to \mathbb{R}$, a function $h = \sum_{e \in E} h_e$, $Z \in B(f)$ such that $\|z - x^*\| \le \epsilon$.
 1: Initialize $\mathcal{S} = \emptyset$
 2: Let $\tilde{F}_1, \tilde{F}_2, \ldots, \tilde{F}_k$ be a partition of $E$ such that $(\nabla h(z))_e = \tilde{c}_i \,\forall e \in F_i$ and $\tilde{c}_i < \tilde{c}_l$ for $i < l$.
 3: **for** $j \in [k-1]$ **do**
 4:     **If** $\tilde{c}_{j+1} - \tilde{c}_j > 2\epsilon$, **then** $\mathcal{S} = \mathcal{S} \cup \{F_1 \cup \cdots \cup F_j\}$        ▷ we discovered a tight set at $x^*$
 5: **end for**
**Return:** $\mathcal{S}$

---

Next we present our INFER2 our Combinatorial relaxed rounding RELAX (**T5**). This tool allows for early termination of iterative algorithms used to compute the projections by checking if we have found all the tight sets at the optimal. Recall that if we find all the tight sets at the optimal solution we can compute the exact projection (using Theorem 1 for example).

---

**Algorithm 4** Combinatorial relaxed rounding (**T5**): RELAX$(\mathcal{S}, \mathcal{V})$

---

**Input:** Submodular function $f : 2^E \to \mathbb{R}$, a function $h = \sum_{e \in E} h_e$, a chain of tight sets $\mathcal{S} = \{S_1, \ldots, S_k\}$
    where $S_1 \subset \cdots \subset S_k = E$, and a set of vertices $\mathcal{V} = \{v_1, \ldots, v_l\}$ where $v_i$ is a vertex of $B(f)$.
 1: Initialize $Flag = False$
 2: Let $\tilde{x} := \arg\min\{h(x) \mid x(S) = f(S) \,\forall S \in \mathcal{S}\}$        ▷ could be solved using Theorem 1
 3: **If** $\tilde{x} \in \text{Conv}(\mathcal{V})$, **then** $Flag = True$        ▷ we guessed optimal solution: $\tilde{x} = x^*$
**Return:** $\tilde{x}$, Flag

---

We now present our second rounding tool ROUND for base polytopes of integral submodular functions, which is algebraic in nature. It only requires a guarantee that the approximate projection be within a (Euclidean) distance of $1/(2|E|^2)$ to the optimal for Euclidean projections and more importantly doesn't depend on knowing the tight sets at the optimal solution. This rounding algorithm runs in time $O(n^2 \log n)$ and is given below.

---

**Algorithm 5** Integer-function rounding (**T6**): ROUND$(\mathcal{S}, \mathcal{V})$

---

**Input:** Submodular function $f : 2^E \to \mathbb{Z}$, a point $y \in \mathbb{Z}^E$, $x \in B(f)$ such that $|x_e - x_e^*| < \frac{1}{2|E|^2}$ for all
    $e \in E$, where $x^* = \Pi_{\mathcal{P}}(y)$ is the Euclidean projection of $y$ on $\mathcal{P}$.
 1: **for** each $e \in E$ **do**
 2:     $z^{(i)} := \arg\min_{s \in \frac{1}{i}\mathbb{Z}} |s - x_e|$, for each $i \in \{1, \ldots, |E|\}$.
 3:     $z_e := \min_i z^{(i)}$
 4: **end for**
 5: **Return** $z$

---

Finally, we present the pseudocode for $AFW$-$update$ used within our A$^2$FW algorithm, which performs an AFW descent step and returns the new iterate along with its active set.

---

**Algorithm 6** Away-steps Frank-Wolfe update ($AFW$-$update(z, v, a, \mathcal{A})$)

---

**Input:** Submodular $f : 2^E \to \mathbb{R}$, convex function $h : B(f) \to \mathbb{R}$, $z \in B(f)$ with active set $\mathcal{A}$, FW vertex
    $v \in B(f)$, and away vertex $a \in B(f)$.
 1: Define the FW gap $g^{\text{FW}} := \langle -\nabla h(z), v - z \rangle$.
 2: **if** $g^{\text{A}} := \langle -\nabla h(z), z - a \rangle \le g^{\text{FW}}$ **then**        ▷ FW gap v/s away gap
 3:     $d := v - z$ and $\gamma^{\text{max}} := 1$.        ▷ choose FW direction
 4: **else**
 5:     $d := z - a$ and $\gamma_t^{\text{max}} := \lambda_a/(1 - \lambda_a)$.        ▷ choose away direction
 6: **end if**
 7: Let $z^+ := z + \gamma d$ for $\gamma = \arg\min_{\gamma \in [0, \gamma_{\text{max}}]} h(z + \gamma d)$
 8: Update $\lambda_v$ for all $v \in \mathcal{A}$ and $\mathcal{A}^+ = \{v \in B(f) \mid \lambda_v > 0\}$        ▷ update active set
**Return:** $z^+, \mathcal{A}^+, g^{\text{FW}}$

---

## C  Missing proofs in Section 3 and the PAV Algortihm

We extend the proof of Lim and Wright [38] and prove Theorem 2. To do that we need some more preliminaries. Consider any strictly convex and continuously differentiable separable function $h : \mathcal{D} \to \mathbb{R}$, defined over a convex set $\mathcal{D}$ such that $B(f) \cap \mathcal{D} \neq \emptyset$ and $\nabla h(\mathcal{D}) = \mathbb{R}^{E}$ (this condition is not restrictive). Recall that the Fenchel-conjugate of $h$, that is $h^*(y) = \sup_{x \in \mathcal{D}} \{\langle y, x \rangle - h(x)\}$ for any $y \in \mathcal{D}^*$. The subdifferential of $h$, i.e. the set of all subgradients of $h$, is defined by $\partial h = \{g \in \mathcal{D}^* : h(y) \geq h(x) + \langle g, y - x \rangle \ \forall \, y \in \mathcal{D}\}$. Since $h$ is strictly convex and differentiable, the subdifferential is unique and given by $\partial h(x) = \nabla h(x)$ for all $x \in \mathcal{D}$. The conjugate subgradient theorem states that for any $x \in \mathcal{D}$, $y \in \mathcal{D}^*$, we have $\partial h(x) = \arg\max_{\tilde{y} \in \mathcal{D}^*} \{\langle x, \tilde{y} \rangle - h^*(\tilde{y})\} = \nabla h(x)$ and $\partial h^*(y) = \arg\max_{\tilde{x} \in \mathcal{D}} \{\langle y, \tilde{x} \rangle - h(\tilde{x})\} = \nabla h^*(y)^{\parallel}$ (see e.g. Corollary 4.21 in [20]). We will need the Fenchel duality theorem, which states that (see e.g. Theorem 4.15 in [20]):

$$\min_{x \in \mathcal{X}} h(x) = \max_{y \in \mathcal{D}^*} -h^*(y) + \min_{x \in \mathcal{X}} y^T x. \tag{4}$$

When $\mathcal{X} = B(f)$, the above result coincides with Proposition 8.1 in [18].

### C.1  Proof of Theorem 2

We first show the following result about minimizing strictly convex functions over polytopes, which states that if we know the optimal (minimal) face, then we can restrict the optimization to that optimal face.

**Lemma 4** (Reduction of optimization problem to optimal face). *Consider any strictly convex function $h : \mathcal{D} \to \mathbb{R}$. Let $\mathcal{P} = \{x \in \mathbb{R}^n : \langle a_i, x \rangle \leq b_i \ \forall \ i \in [m]\}$ be a polytope and assume that $\mathcal{D} \cap \mathcal{P} \neq \emptyset$. Let $x^* = \arg\min_{x \in P} h(x)$, where uniqueness of the optimal solution follows from the strict convexity of $f$. Further, let $I(x^*)$ denote the index-set of active constraints at $x^*$ and $\tilde{x} = \arg\min_{x \in \mathbb{R}^n} \{h(x) \mid \mathbf{A}_{I(x^*)} x \leq \mathbf{b}_{I(x^*)}\}$. Then, we have that $x^* = \tilde{x}^{**}$.*

*Proof.* Let $J(x^*)$ denote the index set of inactive constraints at $x^*$. We assume that $J(x^*) \neq \emptyset$, since otherwise the result follows trivially. Now, suppose for a contradiction that $x^* \neq \tilde{x}$. Due to uniqueness of the minimizer of the strictly convex function over $\mathcal{P}$, we have that $\tilde{x} \notin \mathcal{P}$ (otherwise it contradicts optimality of $x^*$ over $\mathcal{P}$). We now construct a point $y \in \mathcal{P}$ that is a strict convex combination of $\tilde{x}$ and $x^*$ and satisfies $f(y) < f(x^*)$, which contradicts the optimality of $x^*$. To that end, define

$$\gamma := \min_{\substack{j \in J(x^*): \\ \langle a_j, \tilde{x} - x^* \rangle > 0}} \frac{b_j - \langle a_j, x^* \rangle}{\langle a_j, \tilde{x} - x^* \rangle} > 0, \tag{5}$$

with the convention that $\gamma = \infty$ if the feasible set of (5) is empty, i.e. $\langle a_j, \tilde{x} - x^* \rangle \leq 0$ for all $j \in J(x^*)$. Select $\tilde{\theta} \in (0, \min\{\gamma, 1\})$. Further, define $y := x^* + \tilde{\theta}(\tilde{x} - x^*) \neq x^*$ to be a strict convex combination of $x^*$ and $\tilde{x}$. We claim that that $(i)$ $y \in \mathcal{P}$ and $(ii)$ $f(y) < f(x^*)$, which completes our contradiction argument:

$(i)$ *We show that $y \in \mathcal{P}$.* Since all the tight constraints $I(x^*)$ are satisfied at $y$ by construction, to show the feasibility of $y$ we just have to verify that any constraint $j \in J(x^*)$ such that $\langle a_j, \tilde{x} \rangle > b_j > \langle a_j, x^* \rangle$ is feasible at $y$. Indeed, we have

$$\langle a_j, y \rangle = \langle a_j, x^* \rangle + \tilde{\theta} \langle a_j, \tilde{x} - x^* \rangle \leq \langle a_j, x^* \rangle + \gamma \langle a_j, \tilde{x} - x^* \rangle$$
$$\leq \langle a_j, x^* \rangle + b_j - \langle a_j, x^* \rangle = b_j,$$

where we used the fact that $\tilde{\theta} \leq \gamma$ in the first inequality, and the definition of $\gamma$ (5) in the second inequality. This establishes the feasibility of $y \in \mathcal{P}$.

$(ii)$ *We show that $f(y) < f(x^*)$.* Observe that $f(\tilde{x}) \leq f(x^*)$ by construction. Since, $x^* \neq \tilde{x}$, We can now complete the proof of this claim as follows:

$$f(y) = f((1 - \tilde{\theta})x^* + \tilde{\theta}\tilde{x}) < (1 - \tilde{\theta})f(x^*) + \tilde{\theta}f(\tilde{x}) \leq f(x^*),$$

---

$^{\parallel}h^*$ is differentiable since $h$ is strictly convex (see Theorem 26.3 in [61]).

$^{**}$The exact same proof can be used to show that when $\tilde{x}$ is instead defined by $\tilde{x} := \arg\min_{x \in \mathbb{R}^n} \{h(x) \mid \mathbf{A}_{I(x^*)} x = \mathbf{b}_{I(x^*)}\}$ (so that we relax the equalities to inequalities in the definition of $\tilde{x}$), we also have $x^* = \tilde{x}$.

where we used the fact $\tilde{\theta} \in (0,1)$ and the fact that $f$ is strictly convex in the first inequality, and the fact that $f(\tilde{x}) \leq f(x^*)$ in second.

This completes the proof. $\qquad\square$

We also need the following, which lemma shows that the ordering of the optimal solution is the same as the ordering of elements in $y$.

**Lemma 5** (Lemma 1 in [62]). *Let $f : 2^E \to \mathbb{R}$ be any cardinality-based submodular function, that is $f(S) = g(|S|)$ function for some nondecreasing concave function g. Let $\phi : \mathcal{D} \to \mathbb{R}$ be a strictly convex and uniformly separable mirror map where $B(f) \cap \mathcal{D} \neq \emptyset$. Let $x^* := \arg\min_{x \in B(f)} D_\phi(x, y)$ be the Bregman projection of y. Assume that $y_1 \geq \cdots \geq y_n$. Then, it holds that $x_1^* \geq \cdots \geq x_n^*$.*

*Proof.* Suppose on the contrary that $x_i^* < x_j^*$ for $i < j$. Let $\tilde{x}$ be the point obtained by exchanging $x_i^*$ and $x_j^*$. Then, by definition, we have $\tilde{x}$ is feasible in $B(f)$. Moreover,

$$
\begin{aligned}
D_\phi(x^*, y) - D_\phi(\tilde{x}, y) &= \phi(x_i^*) - \phi(y_i) - (\nabla\phi(y))_i(x_i^* - y_i) - \phi(x_j^*) + \phi(y_i) + (\nabla\phi(y))_i(x_j^* - y_i) \\
&\quad + \phi(x_j^*) - \phi(y_j) - (\nabla\phi(y))_j(x_j^* - y_j) - \phi(x_i^*) + \phi(y_j) + (\nabla\phi(y))_j(x_i^* - y_j) \\
&= -(\nabla\phi(y))_i(x_i^* - x_j^*) - (\nabla\phi(y))_j(x_j^* - x_i^*) \\
&= (x_j^* - x_i^*)((\nabla\phi(y))_i - (\nabla\phi(y))_j) \\
&> 0,
\end{aligned}
$$

which is a contradiction. $\qquad\square$

We are now ready to prove Theorem 2:

**Theorem 2** (Dual of projection is isotonic optimization). *Let $f : 2^E \to \mathbb{R}$ be a cardinality-based monotone submodular function, that is $f(S) = g(|S|)$ function for some nondecreasing concave function g. Let $c_i := g(i) - g(i-1)$ for all $i \in [E]$. Let $\phi : \mathcal{D} \to \mathbb{R}$ be a strictly convex and uniformly seperable mirror map. Let $B(f) \cap \mathcal{D} \neq \emptyset$ and consider any $y \in \mathbb{R}^n$. Let $\{e_1, \ldots, e_n\}$ be an ordering of the ground set E such that $y_1 \geq \cdots \geq y_n$. Then, the following problems are primal-dual pairs*

$$
(P) \quad
\begin{aligned}
&\min\ D_\phi(x, y) \\
&\text{subject to } x \in B(f)
\end{aligned}
\qquad
(D) \quad
\begin{aligned}
&\max\ -D_\phi^*(z, y) + z^T c \\
&\text{subject to } z_1 \leq \cdots \leq z_n
\end{aligned}.
\tag{1}
$$

*Moreover, from a dual optimal solution $z^*$, we can recover the optimal primal solution $x^*$.*

*Proof.* Consider the problem of computing a Bregman projection of a point $y$ over a cardinality-based submodular polytope

$$
\min\ D_\phi(x, y) \quad \text{subject to } x(S) \leq g(|S|) \quad \forall S \subset E, \quad x(E) = g(|E|).
\tag{6}
$$

Note that since, $y_1 \geq \cdots \geq y_n$, the previous two lemmas imply that we can reduce to problem to the optimal face, which only includes the constraints that can be active under that ordering. That is, problem (6) can be simplified to only have $n$ constraints as opposed to the original problem which had $2^n$ constraints:

$$
\min\ D_\phi(x, y) \quad \text{subject to } \sum_{i=1}^j x_i \leq g(j) \quad \forall j \in [n-1], \quad \sum_{i=1}^n x_i = g(n).
\tag{7}
$$

Let $C$ denote the feasible region of the simplified optimization problem in (7). Then, using the Fenchel duality theorem (4), we have that the following problems are primal-dual pairs:

$$
(P) \quad
\begin{aligned}
&\min\ D_\phi(x, y) \\
&\text{subject to } x \in B(f)
\end{aligned}
\qquad
(D) \quad \max_{z \in \mathbb{R}^n}\ -D_\phi^*(z, y) + \min_{x \in C} \langle z, x \rangle
\tag{8}
$$

Let us now focus on the $\min_{x \in C} \langle z, x \rangle$ term in the dual problem $(D)$ above. If we let $Z_i = z_i - z_{i+1}$ for $i \in [n-1]$ and $Z_n = z_n$, we have $z_i = \sum_{k=i}^{n} Z_k$. Recall that $c_i = g(i) - g(i-1)$ for $i = 1 \ldots, n$ and note that $c_i \le c_{i-1}$ since $g$ is concave. This gives us

$$\langle z, x \rangle = \langle z, c \rangle + \sum_{i=1}^{n} z_i(x_i - c_i) = \langle z, c \rangle + \sum_{i=1}^{n} \left( \sum_{k=i}^{n} Z_k \right)(x_i - c_i)$$

$$= \langle z, c \rangle + \sum_{k=1}^{n} \left( \sum_{i=1}^{k}(x_i - c_i) \right) Z_k \tag{9}$$

If any $Z_k$ is larger than 0 for any $k \in [n-1]$, then we claim that $\min_{x \in C} \langle z, x \rangle = -\infty$. Indeed, we can set $x_i = c_i$ for all $i \notin \{k, k+1\}$, $x_k \to -\infty$ and $x_{k+1} = c_k + c_{k+1} - x_k$, where it is clear that such a solution is feasible in $C$. This means that we require $Z_k \le 0$ for all $k$ (i.e. $z_{i+1} \ge z_i$ for all $i$). Thus, since $\sum_{k=1}^{n} \left( \sum_{i=1}^{k}(x_i - c_i) \right) Z_k \ge 0$ for all $k \in [n]$ (as $x \in C$), it follows that $\min_{x \in C} \langle z, x \rangle = \langle z, c \rangle$ is obtained by setting $x_i = c_i$ for all $i$ in (9). In other words, $\min_{x \in C} \langle z, x \rangle$ is attained by the vertex of $B(f)$ that corresponds to the ordering induced by the chain constraints.

Thus, we accordingly simplify our dual problem to obtain

$$(P) \quad \begin{array}{l} \min \ D_\phi(x, y) \\ \text{subject to } x \in B(f) \end{array} \qquad (D) \quad \begin{array}{l} \max \ -D_\phi^*(z, y) + z^T c \\ \text{subject to } z_1 \le \cdots \le z_n \end{array}.$$

Furthermore, since $z^*$ is the optimal solution $z^*$ to the Fenchel dual $(D)$, we can use the conjugate subgradient theorem (given in the introduction of this section) to recover a primal solution using $\nabla_x D_\phi(x^*, y) = \nabla\phi(x^*) - \nabla\phi(y) = z^*$. $\qquad \square$

## C.2 PAV Algorithm Implementation and Example

We now propose our algorithm, which solves the dual problem and then maps the dual optimal solution to a primal one using Theorem 2. Best. al [54] show that such problems could be solved exactly in $n$ iterations, using a well known algorithm called the Pool Adjacent Violators (PAV) Algorithm in $O(n)$ time (see Theorem 2.5 in [54]). We adapt the algorithm here in Algorithm 7 to solve $(D)$.

The algorithm begins with the finest partition of the ground set $E$ whose blocks are single integers in $[E]$ and an initial solution (that is possibly infeasible and violates the chain constraints). Then, the algorithm successively merges blocks to reduce infeasibility through *pooling* steps, obtaining a new, coarser partition of the ground set $E$ and an infeasible solution $z$, until $z$ becomes dual feasible. The pooling step is composed of solving an unconstrained version of the dual objective function restricted to a set $S$. We denote this operation by $\text{Pool}_{\phi,y,c}(S) := \arg\min_{\gamma \in \mathbb{R}} \sum_{i \in S} D_{\phi_i}^*(\gamma, y_i) + \gamma c_i$, where the solution is unique by the strict convexity of $\phi_i$. We solve for $\gamma$ by setting the derivative to zero to obtain (see [38] for more details):

$$\sum_{i \in S}(\nabla\phi^{-1})(\gamma + \nabla\phi(y_i)) = \sum_{i \in S} c_i. \tag{10}$$

Consider the case when $\phi(x) = \frac{1}{2}\|x\|^2$ so that our Bregman projection becomes a Euclidean projection. In this case, we have $\nabla\phi(x) = x = (\nabla\phi)^{-1}(x)$ and (10) reduces to computing an average: $\text{Pool}_{\phi,y,c}(S) = \sum_{i \in S}(c_i - y_i)/|S|$[††]. On the other hand, when $\phi(x) = x \ln x - x$ so that our Bregman projection becomes the generalized KL-divergence, we have $\text{Pool}_{\phi,y,c}(S) = \ln \frac{\sum_{i \in S} c_i}{\sum_{i \in S} z_i}$. Henceforth, we assume that the $\text{Pool}_{\phi,y,c}$ operation can be done in $O(1)$ time using oracle access (which is a valid assumption for most widely-used mirror maps). We have thus arrived at the following result which gives the correctness and running time of the PAV algorithm:

**Theorem 7.** *Let $f : 2^E \to \mathbb{R}$ be a cardinality-based submodular function, that is $f(S) = g(|S|)$ function for some concave function $g$. Let $\phi : \mathcal{D} \to \mathbb{R}$ be a strictly convex and uniformly seperable mirror map, where $B(f) \cap \mathcal{D} \ne \emptyset$. Then the output of the PAV algorithm (given in Algorithm 7) is $x^* = \arg\min_{x \in B(f)} D_\phi(x, y)$. Moreover, the running-time of the algorithm is $O(n \log n + nEO)$.*

---

[††]When $\phi(x) = \frac{1}{2}\|x\|^2$, problem (D) in (1) is equivalent to $\min_z\{\frac{1}{2}\|z - (c - y)\|^2 \mid z_1 \le \cdots \le z_n\}$, which is an isotonic regression problem.

**Algorithm 7** Pool Adjacent Violators (PAV) Algorithm

---

**Input:** Cardinality-based submodular function $f(S) = g(|S|) : 2^E \to \mathbb{R}$, strictly convex and uniformly separable mirror map $\phi : \mathcal{D} \to \mathbb{R}$ such that $B(f) \cap \phi \neq \emptyset$, and point to be projected $y \in \mathbb{R}^n$ where $y_1 \geq y_2 \geq \cdots \geq y_n$.

1: Initialize $P \leftarrow \{i | i \in [E]\}$ and $z_i \leftarrow \text{Pool}_{\phi,y,c}(i)$ for all $i \in [E]$.
2: **while** $\exists$ indices $i, i+1 \in \mathcal{P}$ where $z_i > z_{i+1}$ **do**
3:     Let $K(i)$ and $K(i+1)$ be the intervals in $\mathcal{P}$ containing indices $i$ and $i+1$ respectively.
4:     Remove $K(i), K(i+1)$ from $\mathcal{P}$ and add $K(i) \cup K(i+1)$.
5:     set $z_{K(i) \cup K(i+1)} \leftarrow \text{Pool}_{\phi,y,c}(K(i) \cup K(i+1))$.         ▷ `see equation (10)`
6: **end while**
7: Set $x^* \leftarrow \nabla\phi^{-1}(z + \nabla\phi(y))$         ▷ `recover primal solution`
**Return:** $x^* = \arg\min_{x \in B(f)} D_\phi(x, y)$.

---

*Proof.* The proof of this result follows from the fact that we need to sort $y$ in Theorem 2 (which could be done in $O(n \log n)$ time) and the fact that the PAV algorithm solves the dual problem exactly in $n$ iterations using Theorem 2.5 by Best. al [54], where each iteration takes $O(1)$ time. This gives a total running time of $O(n \log n + nEO)$.     □

To explain the algorithm further and see it at work, consider the following example. Suppose we want to compute the Euclidean projection of $y = (4.8, 4.6, 2.7)$ onto the 1-simplex defined over the ground set $E = \{1, 2, 3\}$ with the cardinality-based set function $f(S) = \min\{|S|, 1\}$. In this case we have $c_1 := 1$ and $c_i = 0$ for all $i \in \{2, \ldots, n\}$. The PAV algorithm initializes $z^{(0)} = c - y = (-3.8, -4.6, -2.7)$ using (10). Since $z_1 > z_2$, in the first iteration the algorithm will pool the first two coordinates by averaging them to obtain $z^{(1)} = c - y = (-4.2, -4.2, -2.7)$. Now we have $z_1^{(1)} \leq z_2^{(1)} \leq z_3^{(1)}$ and the algorithm terminates. Moreover, we recover the primal optimal solution using $x^* = z^{(1)} + y = (0.6, 0.4, 0)$.

## D   Missing proofs in Section 4

### D.1   Missing proofs in Section 4.1

#### D.1.1   Proof of Theorem 3

**Theorem 3** (Recovering tight sets from previous projections **(T1)**). *Let $f : 2^E \to \mathbb{R}$ be a monotone submodular function with $f(\emptyset) = 0$. Further, let $y$ and $\tilde{y} \in \mathbb{R}^E$ be such that $\|y - \tilde{y}\| \leq \epsilon$, and $x, \tilde{x}$ be the Euclidean projections of $y, \tilde{y}$ on $B(f)$ respectively. Let $F_1, F_2, \ldots, F_k$ be a partition of the ground set $E$ such that $x_e - y_e = c_i$ for all $e \in F_i$ and $c_i < c_l$ for $i < l$. If $c_{j+1} - c_j > 4\epsilon$ for some $j \in [k-1]$, then the set $S = F_1 \cup \cdots \cup F_j$ is also a tight set for $\tilde{x}$, i.e. $\tilde{x}(S) = f(S)$.*

*Proof.* We show a more general result for uniformly separable divergences based on an $L$-smooth and strictly convex mirror map $\phi$, so that the corresponding Bregman projection is nonexpansive, i.e., if $\|y - \tilde{y}\| \leq \epsilon$ then $\|x - \tilde{x}\| \leq \epsilon$. Let $F_1, F_2, ..., F_k$ be a partition of $E$ such that $\nabla D_\phi(x, y)_e = c_i$ for all $e \in F_i$ and $c_i < c_l$ for $i < l$. We now show that if $c_{j+1} - c_j > 4\epsilon L$ for some $j \in [k-1]$, then the set $S = F_1 \cup \ldots \cup F_j$ is also a tight set for $\tilde{x}$. Let $\nabla D_\phi(x, y) = g$ and $\nabla D_\phi(\tilde{x}, \tilde{y}) = \tilde{g}$ for brevity.

Let $e_j, e_{j+1} \in E$ be such that $g(e_j) = c_j$ and $g(e_{j+1}) = c_{j+1}$. Consider the set of elements $S = \{e_1, \ldots, e_k\}$ that have a partial derivative at $x$ of value at most $c_j$, i.e., $S_j = \{e_i | g(e_i) \leq c_j\}$. Let $\tilde{C}_j := \{\tilde{g}(e_i) : e_i \in S_j\}$ and let $\tilde{C} := \{\tilde{g}(e) : e \in E\}$. Then, we will show every element of the set $\tilde{C}_j$ is smaller than every element of the set $\tilde{C} \setminus \tilde{C}_j$, by showing that $\max \tilde{C}_j \leq \min \tilde{C} \setminus \tilde{C}_j$.

For any $e \in E$, consider $i$ such that $g(e) = c_i$. Then,

$$
\begin{aligned}
|\tilde{g}(e) - c_i| &= |\tilde{g}(e) - g(e)| \\
&\leq \|\tilde{g} - g\|_\infty \\
&\leq \|\tilde{g} - g\|_2 \\
&= \|(\nabla\phi(\tilde{x}) - \nabla\phi(\tilde{y})) - (\nabla\phi(x) - \nabla\phi(y))\|_2
\end{aligned}
$$

$$\leq \|\nabla\phi(\tilde{x}) - \nabla\phi(x)\|_2 + \|\nabla\phi(y) - \nabla\phi(\tilde{y})\|_2$$
$$\leq L\|\tilde{x} - x\|_2 + L\|\tilde{y} - y\|_2$$
$$< 2L\epsilon.$$

We use the result on gradient of Bregman projections for the first equality. The third inequality uses the triangle inequality, the fourth inequality uses $L$-smoothness, and the fifth inequality uses the non-expansiveness of the Euclidean (or Bregman) projection.

Therefore, if $e$ is such that $g(e) = c_i \leq c_j$,

$$\tilde{g}(e) < c_i + 2L\epsilon \leq c_j + 2L\epsilon < c_{j+1} - 2L\epsilon < \tilde{g}(e_{j+1}).$$

The first and last inequalities follow from the inequality we established above, and the second and third inequalities follow by assumption. Similarly, if $i$ is such that $c_i > c_j$, then $\tilde{g}(e) \geq \tilde{g}(e_{j+1})$.

This implies the following: every element of the set $\tilde{C}_j = \{\tilde{g}(e) : e \in S\}$ is smaller than every element of $\tilde{C} \setminus \tilde{C}_j$ as claimed. Since $\phi$ is $L$-smooth (and thus continuously differentiable) and strictly convex, the result then follows using Theorem 1. $\qquad\square$

### D.1.2 Proof of Theorem 4

**Theorem 4** (Adaptively inferring the optimal face **(T2)**). *Let $f : 2^E \to \mathbb{R}$ be monotone submodular with $f(\emptyset) = 0$, $h : \mathcal{D} \to \mathbb{R}$ be a strictly convex and $L$-smooth function, where $B(f) \cap \mathcal{D} \neq \emptyset$. Let $x := \arg\min_{z \in B(f)} h(z)$. Consider any $z \in B(f)$ such that $\|z - x\| \leq \epsilon$. Let $\tilde{F}_1, \tilde{F}_2, \ldots, \tilde{F}_k$ be a partition of the ground set $E$ such that $(\nabla h(z))_e = \tilde{c}_i$ for all $e \in \tilde{F}_i$ and $\tilde{c}_i < \tilde{c}_l$ for $i < l$. Suppose $\tilde{c}_{j+1} - \tilde{c}_j > 2L\epsilon$ for some $j \in [k-1]$. Then, $S = F_1 \cup \cdots \cup F_j$ is tight for $x$, i.e. $x(S) = f(S)$.*

*Proof.* The proof of this theorem utilizes the same ideas as those in the proof of Theorem 3. Consider elements $e_j, e_{j+1} \in E$ be such that $\nabla h(z)(e_j) = \tilde{c}_j$ and $\nabla h(z)(e_{j+1}) = \tilde{c}_{j+1}$.

Let $S_j$ be the set of elements at which $z$ has a partial derivative at most $c_j$. Let $C_j$ be the partial derivative values at $x$ at $S_j$, i.e., $C_j := \{\nabla h(x)_e : e \in S_j\}$ and let $C := \{\nabla h(x)_e : e \in E\}$. Then, we'll show that $\max C_j \leq \min C \setminus C_j$.

For each $e \in E$, there is an $i$ such that $\nabla h(z)_e = \tilde{c}_i$. Then, using the $L$-smoothness of $h$ we have

$$|\nabla h(x)(e) - \tilde{c}_i| = |\nabla h(x)(e) - \nabla h(z)(e)| \leq L\|x - z\|_2 < L\epsilon. \tag{11}$$

Therefore, for any $e$ such that $\tilde{c}_i \leq \tilde{c}_j$,

$$\nabla h(x)(e) < \tilde{c}_i + L\epsilon \leq \tilde{c}_j + L\epsilon < \tilde{c}_{j+1} - L\epsilon < \nabla h(x)(e_{j+1}).$$

The first and last inequalities follow from the inequality established above, and the second and third inequalities follow by definition.

Similarly, if $e$ is such that $\tilde{c}_i > \tilde{c}_j$, then $\nabla h(x)(e) \geq \nabla h(x)(e_{j+1})$. Together, these imply the following: every element of the set $C_j = \{\nabla h(x)(e) : e \in C_j\}$ is smaller than every element of $C \setminus C_j$. Since $h$ is $L$-smooth (and thus continuously differentiable) and strictly convex, the result then follows using Theorem 1. $\qquad\square$

### D.2 Missing proofs in Section 4.2

### D.2.1 Proof of Lemma 1

To prove this lemma we first need the following result, which states for any $x$ in a polytope $\mathcal{P}$, a vertex in an active set for $x$ must like on the minimal face containing $x$:

**Lemma 6.** *Let $\mathcal{P} = \{x \in \mathbb{R}^n : \langle a_i, x \rangle \leq b_i \ \forall \ i \in [m]\}$ be a polytope with a vertex set $\text{vert}(\mathcal{P})$. Consider any $x \in P$ and let $F = \{z \in \mathcal{P} : \langle a_i, x \rangle = b_i \ \forall \ i \in I(x)\}$ be the minimal face containing $x$, where $I(x)$ is the index set of active constraints at $x$. Let $\mathcal{A}(x) := \{S : S \subseteq \text{vert}(P) \mid x$ is a proper convex combination of all the elements in $S\}$ be the set of all possible active sets for $x$, and define $\mathcal{A}(x) := \cup_{A \in \mathcal{A}(x)} A$ to be the union of all vertices appearing in any active set for $x$. Then, we claim that $\mathcal{A}(x) = \text{vert}(F)$.*

*Proof.* We first show that $\mathcal{A}(x) \subseteq \text{vert}(F)$. To do that, we claim that any $A \in \mathcal{A}(x)$ must be contained in vert($F$). Indeed, let $A \in \mathcal{A}(x)$ be any active set for $x$ and fix a vertex $y \in A$ arbitrarily. Define $z := \frac{1}{1-\lambda_y} \sum_{v \in A \setminus \{y\}} \lambda_v v \in \text{Conv}(A)$ to be the point obtained by shifting the weight from $y$ to other vertices in $A \setminus \{y\}$. Then, we can write $x = \lambda_y y + (1-\lambda_y)z$. Now, if $\langle a_i, x \rangle = b_i$, then the fact that $\langle a_i, z \rangle \leq b_i$ implies that $\langle a_i, y \rangle = b_i$, so that $y \in \text{vert}(F)$.

To show the reverse inclusion, we claim that any $v \in \text{vert}(F)$ lies in an active set containing $x$. Let $x$ be in the relative interior of its minimal face $F$ (otherwise $x$ is a vertex, and the case is trivial). Let $v \in \text{vert}(F)$ be arbitrary and we will now construct an active set $A \in \mathcal{A}(x)$ containing $v$. Define $\lambda^* := \max\{\lambda \mid x + \lambda(x-v) \in F\}$ to be the maximum movement from $x$ in the direction $x - v$. Note $\lambda^* > 0$ since $x$ is in the relative interior of $F$. Let $z := x + \lambda^*(x-v) \in F$ to be the point obtained by moving maximally from $x$ along the direction $x - v$. Now observe that $(i)$ we can write $x$ as a proper convex combination of $z$ and $v$: $x = \frac{1}{1+\lambda^*}z + \frac{\lambda^*}{1+\lambda^*}v$; $(ii)$ the point $z$ lies in a lower dimensional face $\tilde{F} \subset F$ since it is obtained by line-search for feasibility in $F$. Letting $\tilde{A}$ be any active set for $z$ (where $\tilde{A} \subseteq \text{vert}(\tilde{F})$ by the first part of the proof), we have that $\tilde{A} \cup \{v\}$ is an active set for $x$. $\qquad\square$

We are now ready to prove our lemma:

**Lemma 1** (Reusing active sets **(T3)**). *Let $\mathcal{P} \subseteq \mathbb{R}^n$ be a polytope with vertex set vert($\mathcal{P}$). Let $x$ be the Euclidean projection of some $y \in \mathbb{R}^n$ on $\mathcal{P}$. Let $\mathcal{A} = \{v_1, \ldots, v_k\} \subseteq \text{vert}(\mathcal{P})$ be an active set for $x$, i.e., $x = \sum_{i \in [k]} \lambda_i v_i$ for $\|\lambda\|_1 = 1$ and $\lambda > 0$. Let $F$ be the minimal face of $x$ and $\Delta := \min_{v \in \partial\text{Conv}(\mathcal{A})} \|x - v\|$ be the minimum distance between $x$ and the boundary of $\text{Conv}(\mathcal{A})$. Then, $\mathcal{A}$ is also an active set for the Euclidean projection of any point $\tilde{y} \in \mathbb{B}_\Delta(y) \cap \text{Cone}(F)$, where $\mathbb{B}_\Delta(y) = \{\tilde{y} \in \mathbb{R}^n \mid \|\tilde{y} - y\| \leq \min\{\Delta, \|x - y\|\}\}$ is a closed ball centered at $y$.*

*Proof.* Let $\tilde{y} \in \mathbb{B}_\Delta(y)$ be arbitrary and let $\tilde{x}$ be its Euclidean projection. Further, let $N$ be the normal cone defined by the face $F$, i.e. the cone of all tight constraints at $x$. By the previous lemma we have that $\text{Conv}(\mathcal{A}) \subseteq F$. Using non-expansiveness of projection operator we have that $\|x - \tilde{x}\| \leq \min\{\Delta, \|x - y\|\}$. Moreover, since $\tilde{y} \in \text{Cone}(F)$, it follows that $\tilde{y} - \tilde{x} \in N$ so that $\tilde{x}$ lies in $F$. Thus, since $\|x - \tilde{x}\| \leq \Delta$, we have that $\tilde{x} \in \text{Conv}(\mathcal{A})$. $\qquad\square$

### D.2.2 Proof of Theorem 5

**Theorem 5** (Linear optimization over faces of $B(f)$ **(T4)**). *Let $f : 2^E \to \mathbb{R}$ be a monotone submodular function with $f(\emptyset) = 0$. Further, let $F = \{x \in B(f) \mid x(S_i) = f(S_i) \text{ for } S_i \in \mathcal{S}\}$ be a face of $B(f)$, where $\mathcal{S} = \{S_1, \ldots S_k | S_1 \subseteq S_2 \ldots \subseteq S_k\}$. Then the modified greedy algorithm (Alg. 2) returns $x^* = \arg\max_{x \in F} \langle c, x \rangle$ in $O(n \log n + nEO)$ time.*

*Proof.* Our proof is an extension of the proof of the greedy algorithm by Edmonds [34]. We follow the notation given in Algorithm 2. The linear programming formulation for our problem is:

$$
\begin{aligned}
\max \ & \langle c, x \rangle \\
\text{s.t. } & x(T) \leq h(T) && \forall \, T \subset E, \\
& x(S_i) = h(S_i) && \forall \, S_i \in \mathcal{S}, \\
& x(E) = h(E).
\end{aligned}
\tag{12}
$$

Consider the dual problem to (12):

$$
\begin{aligned}
\min_y \ & \sum_{T \subseteq E} y_T h(T) \\
\text{s.t. } & \sum_{T \ni e_j} y(T) = c(e_j) && \forall \, j \in [1, n], \\
& y_T \geq 0 && \forall \, T \notin \mathcal{S} \cup \{E\}.
\end{aligned}
\tag{13}
$$

Define $U_j := \{e_1, \ldots, e_j\}$ (that is, the first $j$ elements of the order we have induced in the algorithm). Define $y^*$ as:

$$y^*_{U_j} = c(e_j) - c(e_{j+1}) \qquad\qquad \forall\, j \in [1, n-1],$$
$$y^*_{U_n} = c(e_n),$$
$$y^*_T = 0 \qquad\qquad \forall\, T \subseteq E : T \notin \{U_0, \ldots, U_n\}.$$

We will now show that $y^*$ is such that $\sum_{T \subseteq E} y^*_T h(T) = \langle c, x^* \rangle$ (and that $y^*$ and $x^*$ are feasible), so optimality is implied by strong duality.

Note that $\sum_{T \ni e_j} y^*_T = \sum_{\ell \in [j,n]} y^*_{U_\ell} = c(e_j)$. When $T \notin \{U_1, \ldots, U_n\}$, $y^*_T \geq 0$ trivially. For each $j$, when $U_j \notin \mathcal{S}$, $y^*_{U_j} \geq 0$ by definition of the order we have induced on $E$. Therefore $y^*$ is feasible.

The feasibility of $x^*$ is essentially the same as in the proof of the greedy algorithm for $B(f)$. We show that $x^*(T) \leq f(T)$ for all $T \subseteq E$. We use induction on $T$. When $|T| = 0$, $T = \emptyset$ and $x(T) = f(T) = 0$. Assume now that $|T| > 0$, and let $e_j$ be the element of $T$ with the largest index. Then,

$$\begin{aligned}
x(T) &= x(T \setminus \{e_j\}) + x(e_j) \\
&\leq f(T \setminus \{e_j\}) + x(e_j) \\
&= f(T \setminus \{e_j\}) + f(U_j) - f(U_{j-1}) \\
&\leq f(T).
\end{aligned}$$

The first inequality follows from the induction hypothesis, and the last follows by submodularity. The equalities follow by definition. Finally, a straightforward calculation verifies that

$$\sum_{T \subseteq E} y^*_T h(T) = \sum_{i=1}^{n} y_{U_i} h(U_i) = \sum_{i=1}^{n-1} (c(e_j) + c(e_{j+1})) h(U_i) + c(e_j) h(U_n)$$
$$= \sum_{i=1}^{n} c(e_i)(h(U_i) - h(U_{i-1})) = \langle c, x^* \rangle\,,$$

which proves our claim. $\qquad\square$

### D.3 Missing proofs in Section 4.3

#### D.3.1 Proof of Lemma 2

**Lemma 2** (Rounding to optimal face **(T5)**). *Let $f : 2^E \to \mathbb{R}$ be a monotone submodular function with $f(\emptyset) = 0$. Let $h : \mathcal{D} \to \mathbb{R}$ be a strictly convex, where $B(f) \cap \mathcal{D} \neq \emptyset$. Let $x^* := \arg\min_{x \in B(f)} h(x)$, and let $\mathcal{S} = \{S_1, \ldots S_k\}$ contain some of the tight sets at $x^*$, i.e. $x^*(S_i) = f(S_i)$ for all $i \in [k]$. Further, let $\tilde{x} := \arg\min\{h(x) \mid x(S) = f(S)\, \forall S \in \mathcal{S}\}$ be the optimal solution restricted to the face defined by the tight set inequalities corresponding to $\mathcal{S}$. Then, $x^* = \tilde{x}$ iff $\tilde{x}$ is feasible in $B(f)$. In particular, if $\mathcal{S}$ contains all the tight sets at $x^*$, then $x^* = \tilde{x}$.*

*Proof.* Let $\mathcal{S}^*$ be the set of all tight sets at $x^*$. If the optimal face is known, then we can restrict our original optimization problem to that optimal face by Lemma 4, that is $x^* = \arg\min\{h(x) \mid x(S) = h(S)\, \forall S \in \mathcal{S}^*\}$, which proves the last statement of the lemma. Since the feasible region $\{x \mid x(S) = h(S)\, \forall S \in \mathcal{S}\}$ used to obtain $\tilde{x}$ contains the optimal face, i.e. $\{x \mid x(S) = h(S)\, \forall S \in \mathcal{S}^*\} \subseteq \{x \mid x(S) = h(S)\, \forall S \in \mathcal{S}\}$, it follows that $h(\tilde{x}) \leq h(x^*)$. Thus, if $\tilde{x} \in B(f)$, we must have $\tilde{x} = x^*$, otherwise we contradict the optimality of $x^*$ by the strict convexity of $h$. Conversely, if $\tilde{x} = x^*$, then we trivially have $\tilde{x} \in B(f)$. $\qquad\square$

#### D.3.2 Proof of Lemma 3

**Lemma 3** (Combinatorial Integer Rounding Euclidean Projections **(T6)**). *Let $f : 2^E \to \mathbb{Z}\,(|E| = n)$ be a monotone submodular function with $f(\emptyset) = 0$. Consider $y \in \mathbb{Z}^E$ and let $h(x) = \frac{1}{2}\|x - y\|^2$. Let $x^* := \arg\min_{x \in B(f)} h(x)$. Consider any $x \in B(f)$ such that $\|x - x^*\| < \frac{1}{2n^2}$. Define $Q := \mathbb{Z} \cup \frac{1}{2}\mathbb{Z} \cup \ldots \cup \frac{1}{n}\mathbb{Z}$, and for any $r \in \mathbb{R}$, let $q(r) := \arg\min_{s \in Q} |r - s|$. Then, $q(x_e)$ is unique for all $e \in E$, and the optimal solution is given by $x^*_e = q(x_e)$ for all $e \in E$.*

*Proof.* For brevity, denote $|E| = n$. First, if we are given all the tight sets at the optimal solution as defined in Theorem 1, then we can recover the Bregman projection $\arg\min_{x \in B(f)} \sum_e h_e(x_e)$ by solving the following univariate equation:

$$\sum_{e \in F_i} (\nabla h_e)^{-1}(c_i) = f(F_1 \cup \cdots \cup F_i) - f(F_1 \cup \cdots \cup F_{i-1}) \qquad \forall\, i \in [k]. \tag{14}$$

Using equation (14) and noting that $(\nabla h_e)^{-1}(c_i) = c_i + y_e$ for all $e \in F_i$, we have for each $e \in F_i$,

$$x_e = \frac{f(\cup_{j \in [i]} F_i) - f(\cup_{j \in [i-1]}) - y(F_i)}{|F_i|} + y_e.$$

Since $f, y$ are integral, we have $x_e \in Q$ for all $e \in E$. Further, note that

$$\min_{x,y \in Q, x \neq y} |x - y| = \min_{\ell_1, \ell_2 \in [n], k_1 \ell_2 \neq k_2 \ell_1} \left| \frac{k_1}{\ell_1} - \frac{k_2}{\ell_2} \right| = \min_{\ell_1, \ell_2 \in [n], k_1 \ell_2 \neq k_2 \ell_1} \frac{|k_1 \ell_2 - k_2 \ell_1|}{\ell_1 \ell_2} \geq \frac{1}{n^2}.$$

Therefore, there is a unique element of $Q$ that is within a distance of less than $\frac{1}{2n^2}$ from $x_e^*$. But by assumption, we have $|x_e - x_e^*| \leq \|x - x^*\|_2 < \frac{1}{2n^2}$ for all $e \in E$, which implies that $\arg\min_{s \in Q} |x_e - s|$ is singleton, so that the rounding can be done uniquely. Further, note that for all $r \in \mathbb{R}$,

$$\min_{s \in Q} |r - s| = \min_{k \in [n]} \min_{s \in \frac{1}{k}\mathbb{Z}} |r - s| = \min_{k \in [n]} \min_{t \in \mathbb{Z}} |k \cdot r - t|,$$

which implies the correctness of the algorithm. $\qquad\square$

# E  Preliminaries Needed for the Convergence Proofs

Let $\mathcal{P} \subseteq \mathbb{R}^n$ be a polytope and consider the following optimization problem $\min_{x \in \mathcal{P}} h(x)$, where $h : \mathcal{P} \to \mathbb{R}^n$ is $\mu$-strongly convex and $L$-smooth. Let $x^* = \arg\min_{x \in \mathcal{P}} h(x)$ denote the constrained optimal solution. Consider an iterative descent scheme of the form

$$z^{(t+1)} = z^{(t)} + \gamma_t d_t \tag{15}$$

to solve our optimization problem.

**Measuring progress using smoothness.**  Since $h$ is $L$-smooth, it satisfies the following inequality for all $x, y \in \mathcal{P}$ (see e.g. [63])

$$h(y) \leq h(x) + \langle \nabla h(x), y - x \rangle + \frac{L}{2} \|y - x\|^2. \tag{16}$$

To obtain a measure of progress, consider the smoothness inequality (16) applied with $y \leftarrow z^{(t+1)}$ and $x \leftarrow z^{(t)}$:

$$h(z^{(t+1)}) \leq h(z^{(t)}) + \left\langle \nabla h(z^{(t)}), z^{(t+1)} - z^{(t)} \right\rangle + \frac{L}{2} \|z^{(t+1)} - z^{(t)}\|^2 \tag{17}$$

$$= h(z^{(t)}) + \gamma_t \left\langle \nabla h(z^{(t)}), d_t \right\rangle + \frac{L\gamma_t^2}{2} \|d_t\|^2 \tag{18}$$

Let $\gamma_t^{\max} = \max\{\delta \mid x + \delta d_t \in \mathcal{P}\}$. Now consider the step-size $\gamma_{d_t} := \frac{\langle -\nabla h(z^{(t)}), d_t \rangle}{L \|\mathbf{d}_{z^{(t)}}\|^2}$ minimizing the RHS of the inequality above and suppose for now that $\gamma_{d_t} \leq \gamma_t^{\max}$. Then, plugging in $\gamma_{d_t}$ in (18) and rearranging we have

$$h(z^{(t)}) - h(z^{(t+1)}) \geq \frac{\left\langle -\nabla h(z^{(t)}), d_t \right\rangle^2}{2L \|d_t\|^2}. \tag{19}$$

It is important to note that $\gamma_{d_t}$ is not the step-size we obtain from line-search. It is just used as means to lower bound the progress obtained from the line-search step.

**Measuring primal gaps using (strong) convexity.** To prove convergence results for our algorithms, we also need a dual gap bound on $w(z^{(t)}) := w(z^{(t)}) - w(x^*)$. To do this, use the strong convexity of $h$. Since $h$ is $\mu$-strongly convex, it satisfies the following inequality for all $x, y \in \mathcal{P}$

$$h(y) \geq h(x) + \langle \nabla h(x), y - x \rangle + \frac{\mu}{2}\|y - x\|^2. \tag{20}$$

Applying the above inequality with $\mathbf{y} \leftarrow z^{(t)} + \gamma(x^* - z^{(t)})$ and $x \leftarrow z^{(t)}$:

$$h(z^{(t)} + \gamma(x^* - z^{(t)})) - h(z^{(t)}) \geq \gamma \left\langle \nabla h(z^{(t)}), x^* - z^{(t)} \right\rangle + \frac{\mu\gamma^2\|x^* - z^{(t)}\|^2}{2}.$$

The RHS is convex in $\gamma$ and is minimized when $\gamma^* = \frac{\left\langle -\nabla h(z^{(t)}), x^* - z^{(t)} \right\rangle}{\mu\|x^* - z^{(t)}\|^2}$. Plugging $\gamma^*$ in the above expression and re-arranging we obtain

$$h(z^{(t)} + \gamma(x^* - z^{(t)})) - h(x^*) \leq \frac{\left\langle -\nabla h(z^{(t)}), x^* - z^{(t)} \right\rangle^2}{2\mu\|x^* - z^{(t)}\|^2}.$$

As the LHS is independent of $\gamma$, we can set $\gamma = 1$, which gives

$$w(z^{(t)}) := h(z^{(t)}) - h(x^*) \leq \frac{\left\langle -\nabla h(z^{(t)}), x^* - z^{(t)} \right\rangle^2}{2\mu\|x^* - z^{(t)}\|^2}. \tag{21}$$

Further, using Cauchy-Schwartz to bound the right hand side of (21), we can also obtain the following optimality measure, which is known as the *PL-inequality*:

$$w(z^{(t)}) := h(z^{(t)}) - h(x^*) \leq \frac{\|\nabla h(z^{(t)})\|^2}{2\mu}. \tag{22}$$

Another final measure of optimality that we will use is the *Wolfe Gap*:

$$h(z^{(t)}) := h(z^{(t)}) - h(x^*) \leq \left\langle -\nabla h(z^{(t)}), x^* - z^{(t)} \right\rangle \leq \max_{v \in \mathcal{P}} \left\langle -\nabla h(z^{(t)}), v - z^{(t)} \right\rangle. \tag{23}$$

where the first inequality uses the convexity of $h$.

# F    Proof of Theorem 6

## F.1    Proof of Theorem 6

The proof of convergence for $A^2FW$ follows simply from the iteration-wise convergence rate of Lacoste-Julien and Jaggi [6], and properties of convex minimizers over submodular polytopes. Once we detect a tight inequality, we can restrict the feasible region to a smaller face of the polytope. Since this happens only a linear number of times, we get linear convergence with $A^2FW$ as well.

We first recall the definition of the restricted pyramidal width constant:

**Definition 1** (Restricted pyramidal width). *Let $\mathcal{P} \subseteq \mathbb{R}^n$ be a polytope with vertex set $\mathrm{vert}(\mathcal{P})$. Let $F \subseteq \mathcal{P}$ be any face of $\mathcal{P}$. Then, the pyramidal width restricted to to $F$ is defined as*

$$\rho_F := \min_{\substack{F' \in faces(F) \\ x \in F' \\ r \in cone(F'-x)\setminus\{0\}:}} \min_{A \in \mathcal{A}(x)} \max_{v \in F', a \in A} \left\langle \frac{r}{\|r\|}, v - a \right\rangle, \tag{24}$$

*where $\mathcal{A}(x) := \{A \mid A \subseteq \mathrm{vert}(\mathcal{P})$ such that $x$ is a proper convex combination of all the elements in $A\}$.*

To prove Theorem 6, we need the following result:

**Theorem 8.** *Let $\mathcal{P} \subseteq \mathbb{R}^n$ be a polytope. Consider any strongly convex and smooth function $h : \mathcal{P} \to \mathbb{R}$. Further, Consider any suboptimal iterate $z^{(t)}$ of the $A^2FW$ algorithm, and let $\mathcal{A}_t$ be its active set and $K$ be its minimal face. Let $x^* := \arg\min_{x \in \mathcal{P}} h(x)$ and $F$ be a face containing $x^*$ such that $F \supseteq K$. Further, denote $r := -\nabla h(z^{(t)})$ and $\hat{e} := z^{(t)} - x^*/\|z^{(t)} - x^*\|$. Define the pairwise FW direction at iteration $t$ to be $d_t^{PFW} := v^{(t)} - a^{(t)}$, where recall that $v^{(t)} = \arg\max_{v \in F} \left\langle r, v - z^{(t)} \right\rangle$ and $a^{(t)} = \arg\max_{a \in \mathcal{A}_t} \left\langle r, z^{(t)} - a \right\rangle$. Then, we have*

$$\frac{\left\langle r, d_t^{PFW} \right\rangle}{\langle r, \hat{e} \rangle} \geq \rho_F, \tag{25}$$

*where $\rho_F$ is the pyramidal width of $\mathcal{P}$ restricted to $F$ as defined in (24).*

The proof of this result follows by applying Theorem 3 in [6] to the face $F$ instead of the whole polytope (since both $x^*$ and $z^{(t)}$ lie in $F$ and we are doing LO over $F$). We reproduce the proof here for completeness. We need the following lemma from Lacoste-Julien and Jaggi [6] for the proof:

**Lemma 7** (Lemma 5 in [6]). *Let $z$ be at the origin, inside a polytope $\mathcal{P}$ and suppose that $r \in \mathrm{Aff}(\mathcal{P})$ is not a feasible direction for $\mathcal{P}$ from $z$ (i.e. $r \notin \mathrm{Cone}(\mathcal{P})$). Then a feasible direction in $\mathcal{P}$ minimizing the angle with $r$ lies on a facet $F'$ of $\mathcal{P}$ that includes the origin $z$. That is:*

$$\max_{e \in \mathcal{P}} \left\langle r, \frac{e}{\|e\|} \right\rangle = \max_{e \in F'} \left\langle r, \frac{e}{\|e\|} \right\rangle = \max_{e \in F'} \left\langle r', \frac{e}{\|e\|} \right\rangle \tag{26}$$

*where $F'$ contains $z$, and $r'$ is defined as the orthogonal projection of of $r$ on $\mathrm{Aff}(F')$.*

*Proof of Theorem 8.* As $z^{(t)}$ is not optimal, we require that $\langle r, \hat{e} \rangle > 0$. Let $\mathcal{A}(z^{(t)})$ denote all the possible active sets for $z^{(t)}$. Then, we have

$$\left\langle \frac{r}{\|r\|}, d_t^{\mathrm{PFW}} \right\rangle = \max_{v \in F, a \in \mathcal{A}_t} \left\langle \frac{r}{\|r\|}, v - a \right\rangle \geq \min_{A \in \mathcal{A}(z^{(t)})} \max_{v \in F, a \in A} \left\langle \frac{r}{\|r\|}, v - a \right\rangle. \tag{27}$$

By Cauchy-Schwartz, we have $\| \langle r, \hat{e} \rangle \| \leq \|r\|$. First consider the case when $r$ is a feasible direction at $z^{(t)}$, i.e. $r \in \mathrm{Cone}(K - z^{(t)}) \subseteq \mathrm{Cone}(F - z^{(t)})$. Then $r$ appears in the set of directions considered in the definition of the restricted pyramidal width (24) for $F$ and so from (27), we have that the inequality (25) holds.

Now, suppose that $r$ is not feasible for $z^{(t)}$. As $z^{(t)}$ is fixed, we work on the centered face at $z^{(t)}$ to simplify the statements, i.e. let $\tilde{F} := F - z^{(t)}$. Then, we have the following worst-case bound for (25) as $x^* \in F$

$$\frac{\langle r, d_t^{\mathrm{PFW}} \rangle}{\langle r, \hat{e} \rangle} \geq \max_{v \in F, a \in \mathcal{A}_t} \langle r, v - a \rangle \left( \max_{v \in \tilde{F}} \left\langle r, \frac{v}{\|v\|} \right\rangle \right)^{-1}. \tag{28}$$

The first term on the RHS of (28) just comes from the definition of $d_t^{\mathrm{PFW}}$ (with equality), whereas the second term is considering the worst case possibility for $x^*$. Note also that the second term has to be strictly greater to zero since $z^{(t)}$ is not optimal.

Without loss of generality, we can assume that $r \in \mathrm{Aff}(\tilde{F})$. Otherwise we can just project it onto $\mathrm{Aff}(\tilde{F})$ as any orthogonal component would not change the inner products appearing in (28). If (this projected) $r$ is feasible from $z^{(t)}$, then we again have the lower bound (27) arising in the definition of the restricted pyramidal width. We thus assume that $r$ is not feasible.

By Lemma 7, we have we have the existence of a facet $F'$ of $\tilde{F}$ that includes the origin $z^{(t)}$ such that:

$$\max_{e \in \tilde{F}} \left\langle r, \frac{e}{\|e\|} \right\rangle = \max_{e \in F'} \left\langle r, \frac{e}{\|e\|} \right\rangle = \max_{e \in F'} \left\langle r', \frac{e}{\|e\|} \right\rangle. \tag{29}$$

Let us now look at how the numerator of (28) transforms when considering $r'$ and $F'$:

$$\max_{v \in F, a \in \mathcal{A}_t} \langle r, v - a \rangle = \max_{v \in F} \left\langle r, v - z^{(t)} \right\rangle + \max_{a \in \mathcal{A}_t} \left\langle -r, a - z^{(t)} \right\rangle \tag{30}$$

$$\geq \max_{v \in F \cap (F' + z^{(t)})} \left\langle r, v - z^{(t)} \right\rangle + \max_{a \in \mathcal{A}_t \cap (F' + z^{(t)})} \left\langle -r, a - z^{(t)} \right\rangle \tag{31}$$

$$= \max_{v \in (F' + z^{(t)})} \left\langle r', v - z^{(t)} \right\rangle + \max_{a \in \mathcal{A}_t} \left\langle -r', a - z^{(t)} \right\rangle \tag{32}$$

$$= \max_{v \in (F' + z^{(t)}), a \in \mathcal{A}_t} \langle r', v - a \rangle \tag{33}$$

where in (31) we used the fact that $(F' + z^{(t)}) \subseteq F$ and $(\mathcal{A}_t - z^{(t)}) \subseteq \mathcal{K}$ for any face $\mathcal{K}$ of $\tilde{F}$ containing the origin $z^{(t)}$. Thus $\mathcal{A}_t = \mathcal{A}_t \cap (F' + z^{(t)})$, and the second term on the first line actually yields an equality for the second line. In (32) we used the fact that The $r - r'$ is orthogonal to members $F'$, as $r'$ is obtained by orthogonal projection.

Now plugging (28) into (33) we have:

$$\frac{\langle r, d_t^{\mathrm{PFW}} \rangle}{\langle r, \hat{e} \rangle} \geq \max_{v \in (F' + z^{(t)}), a \in \mathcal{A}_t} \langle r', v - a \rangle \left( \max_{v \in F'} \left\langle r', \frac{v}{\|v\|} \right\rangle \right)^{-1}, \tag{34}$$

and we are back to a similar situation to (28), with the lower dimensional $F'$ playing the role of the polytope $\tilde{F}$, and $r' \in \text{Aff}(F')$ playing the role of $r$. If $r'$ is feasible from $z^{(t)}$ in $F'$, then $r'$ and the lower dimensional face $(F' + z^{(t)})$ appear in the set of directions considered in the definition of the restricted pyramidal width (24) (note that we have $(F' + z^{(t)})$ as $F'$ is a face of the centered face $\tilde{F}$)

Otherwise (if $r' \notin \text{Cone}(F'))$, then we can repeat the above process to obtain a new direction $r''$ and lower dimensional face $F''$ such that we can repeat the steps in (29) - (34). We again check if $r''$ is feasible from $z^{(t)}$ in $F''$. If not, we keep repeating the above process as long as we do not get a feasible direction. This process must stop at some point; ultimately, we will reach the lowest dimensional face $K$ that contains $z^{(t)}$. As $z^{(t)}$ lies in the relative interior of $K$, then all directions in $\text{Aff}(K)$ are feasible, and so the projected $r$ will have to be feasible. Moreover, by stringing together the equalities of the type (29) for all the projected directions, we know that $\max_{e \in K} \left\langle r_{final}, \frac{e}{\|e\|} \right\rangle > 0$ (as we originally had $\langle r, \hat{e} \rangle > 0$), and thus $K$ is at least one-dimensional and we also have $r_{final} \neq 0$ (this last condition is crucial to avoid having a lower bound of zero!). $\qquad\square$

We are now ready to prove our convergence theorem:

**Theorem 6** (Convergence rate of A$^2$FW)**.** *Let $f : 2^E \to \mathbb{R}$ be a monotone submodular function with $f(\emptyset) = 0$ and $f$ monotone. Consider any smooth strongly convex function $h(\cdot)$ with unique optimal $x^* \in B(f)$. Let $\mathcal{S}$ be the tight sets found up to iteration $t$ and $F(\mathcal{S})$ be the face defined by these tight sets. Then, the primal gap $w(z^{(t+1)}) := h(z^{(t+1)}) - h(x^*)$ of A$^2$FW decreases geometrically at each step that is not a drop step$^{\ddagger\ddagger}$ nor a restart step:*

$$w(z^{(t+1)}) \leq \left( 1 - \frac{\mu \rho_{F(\mathcal{S})}^2}{4LD^2} \right) w(z^{(t)}), \text{where } D \text{ is the diameter of } B(f) \text{ and} \qquad (3)$$

$\rho_{F(\mathcal{S})}$ *is the pyramidal width of $B(f)$ restricted to $F(\mathcal{S})$ (as defined by (24)). Moreover, in the worst case, the number of iterations to get an $\epsilon$-accurate solution is $O\left( (nLD^2/(\mu \rho_{B(f)})^2) \log(1/\epsilon) \right)$.*

*Proof.* Recall that in the A$^2$FWwe either take the FW direction $d_t = v^{(t)} - z^{(t)}$ or the away direction $d_t = z^{(t)} - a^{(t)}$ depending on which direction has a higher inner product with $-\nabla h(z^{(t)})$. Defining $d_t^{\text{PFW}} := v^{(t)} - a^{(t)}$ to be the *pairwise* FW direction, this implies the following key inequality

$$2 \left\langle -\nabla h(z^{(t)}), d_t \right\rangle \geq \left\langle -\nabla h(z^{(t)}), v^{(t)} - z^{(t)} \right\rangle + \left\langle -\nabla h(z^{(t)}), z^{(t)} - a^{(t)} \right\rangle = \left\langle -\nabla h(z^{(t)}), d_t^{\text{PFW}} \right\rangle.$$
$$(35)$$

We proceed by cases depending on whether the step size chosen by line search is maximal or not, i.e. whether $\gamma_t = \gamma_t^{\max}$ or not:

**Case 1:** *The step size evaluated from line-search is not maximal, i.e.$\gamma_t < \gamma_t^{\max}$ so that we have 'good' step.* Recall from Section E that $\gamma_{d_t} = \frac{\langle -\nabla h(x_t), d_t \rangle}{L \|d_{x_t}\|^2}$ is the step size obtained from optimizing the smoothness inequality to obtain (19). We claim that we can use the step size from $\gamma_{d_t}$ to lower bound the progress even if $\gamma_{d_t}$ is not a feasible step size (i.e. when $\gamma_{d_t} > 1$). To see this, note that the optimal solution of the line-search step is in the interior of the interval $[0, \gamma_t^{\max}]$. Define $x_\gamma := z^{(t)} + \gamma d_t$. Then, because $h(x_\gamma)$ is convex in $\gamma$, we know that $\min_{\gamma \in [0, \gamma_t^{\max}]} h(x_\gamma) = \min_{\gamma \geq 0} h(x_\gamma)$ and thus $\min_{\gamma \in [0, \gamma_t^{\max}]} h(x_\gamma) = h(z^{(t+1)}) \leq h(x_\gamma)$ for all $\gamma \geq 0$. In particular, $h(z^{(t+1)}) \leq h(x_{\gamma_{d_t}})$. Hence, we can use (19) to bound the progress

---

$^{\ddagger\ddagger}$A drop step is when we take an away step with a maximal step size so that we drop a vertex from the current active set.

per iteration as follows:

$$w(z^{(t)}) - w(z^{(t+1)}) = f(z^{(t)}) - f(z^{(t+1)})$$

$$\geq \frac{\left\langle -\nabla h(z^{(t)}), d_t \right\rangle^2}{2L\|d_t\|^2} \tag{36}$$

$$\geq \frac{\left\langle -\nabla h(z^{(t)}), d_t \right\rangle^2}{2LD^2} \tag{37}$$

$$\geq \frac{\left\langle -\nabla h(z^{(t)}), d_t^{\mathrm{PW}} \right\rangle^2}{8LD^2} \tag{38}$$

$$\geq \frac{\rho_{F(\mathcal{S})}}{8LD^2} \frac{\left\langle -\nabla h(z^{(t)}), x^* - z^{(t)} \right\rangle^2}{\|x^* - z^{(t)}\|^2} \tag{39}$$

$$\geq \left(\frac{\rho_{F(\mathcal{S})}}{D}\right)^2 \frac{\mu}{4L} w(z^{(t)}) \qquad . \tag{40}$$

We used the optimized smoothness inequality (19) in (36). The inequality in (38) uses our key pairwise inequality (35). In (39), we used the fact that $x^*, z^{(t)} \in F(\mathcal{S})$ by construction since $t$ is not a rounding iteration and (drop) away steps can only take us to lower dimensional faces of $F(\mathcal{S})$ by Lemma 6, and thus we can apply Theorem 8 to go from (38) to (39). Finally, (40) follows from the primal gap bound we get via strong convexity (21). This shows the rate stated in the theorem.

**Case 2:** *We have a boundary case:* $\gamma_t = \gamma_t^{\max}$. We further divide this case into two sub-cases:

(a) First assume that $\gamma_t = \gamma_t^{\max}$ and we take a FW step, i.e. $d_t = v^{(t)} - z^{(t)}$ so that $\gamma_t^{\max} = 1$. We can assume that the step size from smoothness $\gamma_{d_t}$ is not feasible, i.e. $\gamma_{d_t} > \gamma_t^{\max}$ since otherwise we can use using same argument as above in Case 1 to again obtain a $\left(1 - \left(\frac{\rho_{F(\mathcal{S})}}{D}\right)^2 \frac{\mu}{4L}\right)$-geometric rate of decrease. Now, observe that $\gamma_{d_t} = \frac{\left\langle -\nabla h(z^{(t)}), d_t \right\rangle}{L\|d_t\|^2} > \gamma_t^{\max} = 1$ implies that $\left\langle -\nabla h(z^{(t)}), d_t \right\rangle \geq L\|d_t\|_2^2$. Hence, using the fact that $\gamma_{d_t} > \gamma_t^{\max} = 1$ in the smoothness inequality in (18), we have

$$h(z^{(t)}) - h(z^{(t+1)}) \geq \left\langle -\nabla h(z^{(t)}), d_t \right\rangle - \frac{L}{2}\|d_t\|_2^2$$

$$\geq \frac{\left\langle -\nabla h(z^{(t)}), d_t \right\rangle}{2} \qquad \text{(using } \gamma_t > \gamma_{d_t}^{\max} = 1)$$

$$\geq \frac{h(z^{(t)})}{2} \qquad \text{(using Wolfe gap (23))}$$

Hence, we get a geometric rate of decrease of 1/2.

(b) Finally, assume that $\gamma_t = \gamma_t^{\max}$ and we take an away step, i.e. $d_t = z^{(t)} - a^{(t)}$. In this case (for which we cannot show progress) we will show that these drop steps can happen at most $t/2$ times up to iteration $t$, and hence the bound on the good-steps in the theorem statement. Let $Add_t$ be the number of steps that added a vertex in the active set (only standard FW steps can do this) and let $Drop_t$ be the number of drop steps upto iteration $t$. Then, we have that $|\mathcal{A}_t| = |\mathcal{A}_0| + Add_t - Drop_t$. Moreover, we have that $Add_t + Drop_t \leq t$. We thus have $1 \leq |\mathcal{A}_t| \leq |\mathcal{A}_0| + t - 2Drop_t$, implying that $Drop_t \leq \frac{t}{2}$.

Note that $(i)$ $\rho_{B(f)} \leq \rho_{F(\mathcal{S})}$ for any chain $\mathcal{S}$ since $F(\mathcal{S}) \subseteq B(f)$; $(ii)$ anytime we restart the algorithm, we do so at a vertex of $B(f)$ and thus the increase in the primal gap resulting from the restart is bounded as $h$ is finite over $B(f)$. Thus, since $Drop_t \leq \frac{t}{2}$, and the number of rounding steps is at most $n$ (as the length of any chain of tight sets at $x^*$ is at most $n$), we have that the number of iterations to get an $\epsilon$-accurate solution is $O\left(n\frac{L}{\mu}\left(\frac{D}{\rho_{B(f)}}\right)^2 \log \frac{1}{\epsilon}\right)$ in the worst case. This concludes the proof. $\qquad\square$

# G   Computations

We implemented all algorithms in Python 3.5+, utilizing numpy and scipy for some of our functions. We used these packages from the Anaconda 4.7.12 distribution as well as Gurobi 9 [64] as a black box solver for some of the oracles assumed in the paper. The first experiment was performed on a 16-core machine with Intel Core i7-6600U 2.6-GHz CPU and 256GB of main memory. The second experiment was performed by reserving 5 GB of memory for each run of the experiment on a 24-core Linux x86-64 machines[§§].

**First experiment: Tight cuts.**   We consider $m = 500$ random points $y_1, \ldots, y_m$ obtained by perturbing a random $y_0 \in \mathbb{R}^{100}$ (where $y_0$ is itself sampled from a multivariate Gaussian distribution with mean 100, standard deviation 100) using multivariate Gaussian noise with mean zero and standard deviation $\epsilon = 1/50$. We compute the Euclidean projections of $y_0, y_1, \ldots, y_m$ (exactly) over the permutahedron. The results are plotted in Figure 4-left. Let $\mathcal{S}_i \subseteq 2^E$ represent the chain of tight sets for the projection of point $y_i$, where $E = \{e_1, \ldots, e_{100}\}$ is the ground set. The fraction of tight inequalities for each point $y_i$ that were already tight for some other previous point $y_0, \ldots, y_{i-1}$. The tight sets for the projection of $y_i$ that were also tight for a previous point in $y_1, \ldots, y_{i-1}$ is then $\left| \mathcal{S}_i \cap \left( \bigcup_{j \in [i-1]} \mathcal{S}_j \right) \right|$. The green plot is a cumulative plot of the fraction of tight sets previously seen, that is, it plots

$$\frac{\sum_{i \in [k]} \left| \mathcal{S}_i \cap \left( \bigcup_{j \in [i-1]} \mathcal{S}_j \right) \right|}{\sum_{i \in [k]} |\mathcal{S}_i|}$$

against $k$, the number of points projected so far.

Let $t_i$ be the number of tight sets in $\mathcal{S}_i$ inferred by using Theorem 3 using the projections of $y_1, \ldots, y_{i-1}$; note that $t_i \leq |\mathcal{S}_i \cap (\cup_{j \in [i-1]} \mathcal{S}_j)|$. The blue line plots

$$\frac{\sum_{i \in [k]} t_i}{\sum_{i \in [k]} |\mathcal{S}_i|}$$

against $k$. The plot lines themselves average over 500 independent runs of this experiment, while the shaded region is a 15-85 percentile plot across these runs. Note that our theoretical results give almost tight computational results, that is, we can recover most of the tight sets common between close points using Theorem 3.

**Second experiment: Online learning.**   Next, motivated by the trade-off in regret versus time for online mirror descent (OMD) and online Frank-Wolfe (OFW) variants, we conduct an online convex optimization experiment on the permutahedron (denoted by $B(f)$) with $n = 50$ elements. The loss functions in each iteration are (noisy) linear, and we use (i) Online Frank-Wolfe (OFW) and (ii) Online Mirror Descent (OMD) with the projection subproblem solved using Away-step Frank-Wolfe (AFW) and its variants enhanced by our toolkit.

We consider a time horizon of $T = 1000$, and consider two parameters $a, b$. We consider $a$ random permutations $\sigma_i$ ($i \in [a]$) close within a swap distance of $b$ from each other. We then define loss functions $\ell^{(t)}(x) = \langle c^{(t)}, x \rangle$ for any $x \in B(f)$, where $c^{(t)}$ is the click-through-rate observed when $x$ is played in the learning framework. We construct $c^{(t)}$ randomly as follows: (i) sample a vector $v \sim [0, 1]^n$ uniformly at random, (ii) select a random $\sigma_i$ for $i \in [a]$, and sort $v$ for it to be consistent with $\sigma_i$, that is, $v_{\sigma_i^{-1}(n)} \geq v_{\sigma_i^{-1}(n-1)} \geq \ldots \geq v_{\sigma_i^{-1}(1)}$, and (iii) let $c^{(t)} = v/\|v\|_1$. This $c^{(t)}$ mimics a random click-through-rate close to the random preferences (permutations) in $[a]$. We run our experiment for two settings: (i) $a = 1$, and (ii) $a = 6, b = 6$. For our learning problem, we then run Online Frank Wolfe (OFW) and Online Mirror Descent (OMD) variants with the projection solved by using AFW and the toolkit proposed: (1) OMD-UAFW: OMD with projection using unoptimized away-step Frank-Wolfe, (2) OMD-ASAFW: OMD with projection using AFW with reused active sets, (3) OMD-TSAFW: OMD with projection using AFW with INFER, RESTRICT, and ROUNDING, (4) OMD-A$^2$FW OMD with adaptive AFW, (5) OMD-PAV: OMD with projection using pool adjacent violators, and (6) OFW. We call the first four variants as OMD-AFW variants. In all the AFW variants, we stop and output the solution when the FW gap $g^{FW}$ is at most $\epsilon = 10^{-3}$. The OFW variant we

---

[§§]Performed on the high-performance computing cluster of the Industrial and Systems Engineering department at the Georgia Institute of Technology.

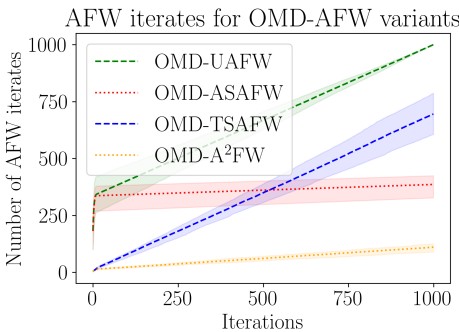 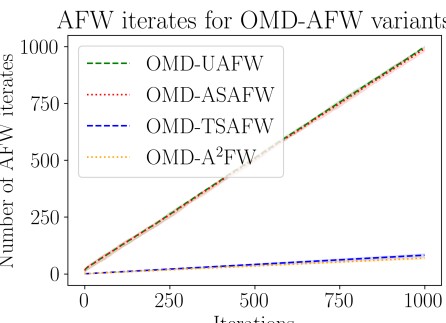

Figure 5: 25-75% percentile plots of number of AFW iterations (cumulative) for OMD-AFW variants over 20 runs for first loss setting (left) and second loss setting (right) for computations in section G

implemented is that of Hazan and Minasyan [7] developed in 2020, which is state-of-the-art and has a regret rate of $O(T^{2/3})$ for smooth and convex loss functions.

As stated previously, we run the experiment 20 times each for (i) $a = 1$ and (ii) $a = 6, b = 6$. Since the run time varied across all runs, we normalized the run time for OMD-UAFW as 1000 (other variants being normalized) in each run to take an equally-weighted average of run times.

Figures 4-middle and 4-right show improvements in run time for OMD-AFW variants, and show significant speed ups of the optimized OMD-AFW variants over OMD-UAFW. Each iteration of OMD involves projecting a point on the permutahedron, and the cumulative run times for these projections are plotted. We remark that OMD-PAV and OFW are much faster than the OMD-AFW variants; however, OMD-PAV suffers from the limitation that it only applies to cardinality-based submodular polytopes, while OFW has significantly higher regret.

The regret for all OMD variants (including OMD-PAV) was observed to be quite similar. Figure 5 shows the total number of iterations of the inner AFW loop for the four OMD-AFW variants plotted cumulatively across the $T = 1000$ projections in the outer OMD loop. AFW for optimized variants that reuse active sets finishes in much fewer AFW iterations over the unoptimized variant, which contributes to a better running time and indicates that we are efficiently reusing information from AFW iterates. These results are summarized in Table 4.

|  | **OMD-AFW Variants** | | | | | |
|  | **UAFW** | **ASAFW** | **TSAFW** | **A$^2$FW** | **OFW** | **OMD-PAV** |
|---|---|---|---|---|---|---|
| **$a = 1$** | | | | | | |
| Regret | 1000 | 1000 | 1016 | 1012 | 520900 | 1000 |
| Runtime | 1000 | 962.3 | 7.372 | 1.306 | 0.03271 | 0.04222 |
| AFW Iterates | 1000 | 386.1 | 695.3 | 110.8 | - | - |
| **$a = 6, b = 6$** | | | | | | |
| Regret | 1000 | 1000 | 1001 | 1002 | 10170 | 1000 |
| Runtime | 1000 | 1014 | 0.9600 | 0.7194 | 0.001852 | 0.002618 |
| AFW Iterates | 1000 | 990.5 | 82.23 | 69.54 | - | - |

Table 4: A comparison of total runtime, regret, and numbers of AFW iterates for computations in Section G averaged over 20 runs of the experiment. The corresponding values for OMD-UAFW are normalized to 1000 and all numbers are reported to 4 significant digits.

## H   Additional computations

We detail some computations on submodular polytopes that are not cardinality-based. We conduct an experiment similar to the online learning experiment in the main body of paper by replacing the underlying submodular base polytope, as described below. We also change the stopping condition for

| | OMD-AFW Variants | | | | |
|---|---|---|---|---|---|
| | **UAFW** | **ASAFW** | **TSAFW** | **A²FW** | **OFW** |
| $a = 1$ | | | | | |
| Regret | 1000 | 1000 | 724 | 723 | 19880 |
| Runtime | 1000 | 728.2 | 400.0 | 84.45 | 10.95 |
| AFW Iterates | 1000 | 204.5 | 935.6 | 147.5 | - |
| $a = 6, b = 6$ | | | | | |
| Regret | 1000 | 1000 | 921.1 | 921.2 | 6584 |
| Runtime | 1000 | 945.2 | 405.7 | 356.7 | 0.4924 |
| AFW Iterates | 1000 | 882.0 | 481.4 | 390.9 | - |

Table 5: A comparison of total runtime, regret, and numbers of AFW iterates for computations in Section H averaged over 20 runs of the experiment. The corresponding values for OMD-UAFW are normalized to 1000 and all numbers are reported to 4 significant digits.

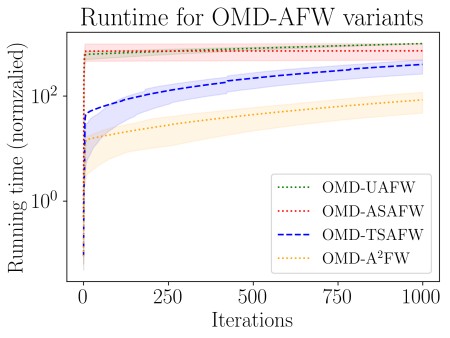 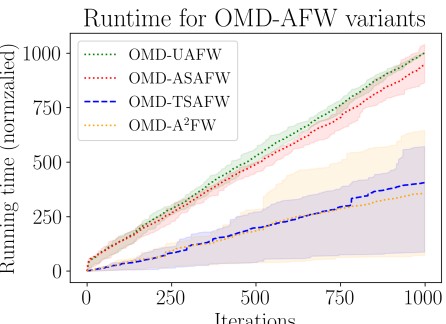

Figure 6: 25-75% percentile plots of runtime for OMD-AFW variants over 20 runs for first loss setting (left) and second loss setting (right) for computations in section H.

the AFW variants, we stop and output the solution when the FW gap $g^{FW}$ is lower than $\epsilon = 10^{-4}$, or if the algorithm rounds the point to the exact solution.

We consider $n = 50$ elements in the ground set and build a submodular function $f : 2^n \to \mathbb{R}$. For a parameter $p \in [0, 1]$, create a random bipartite graph $G$ with bipartition $(U, V)$, where $U = V = [n]$ and each edge $uv, u \in U, v \in V$ is present independently with probability $p$. For each $T \subseteq U$, $f(T)$ is the number of neighbors of $T$ in $V$, that is, $f(T) = |\{v \in V : (u, v) \in E(G) \text{ for some } u \in T\}|$. It can be shown that $f$ is submodular and is not cardinality-based in general. We fix $p = 0.2$ in our case.

The loss functions are generated in the same way as for the online learning setup, and likewise we consider two setups: (i) $a = 1$ and (ii) $a = 6, b = 6$. We do not consider OMD-PAV variant in this experiment because the PAV algorithm is restricted to cardinality-based submodular polytopes.

Figure 6 shows significant speed ups of the optimized OMD-AFW variants over OMD-UAFW for $a = 1$ and for $a = 6, b = 6$. We remark that OFW is much faster than the OMD-AFW variants; however, it has significantly higher regret (on average, 20 to 30 times as much as OMD-AFW variants for $a = 1$ and 6 to 7 times as much as OMD-AFW variants for $a = 6, b = 6$). Figure 7 shows mild improvements in regret for OMD-A²FW over OMD-UAFW. This improvement in regret arises from our rounding procedure: AFW outputs only an approximate solution to the problem (depending on the FW gap stopping threshold $\epsilon$) but A²FW can potentially round to the exact solution, resulting in lower regret. These results are summarized in Table 5.

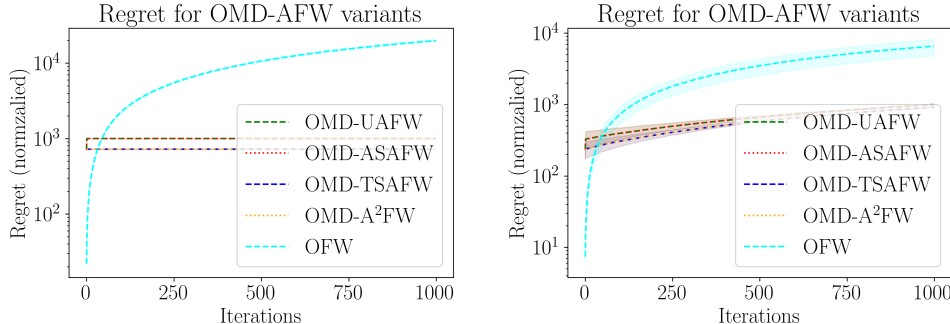

Figure 7: 25-75% percentile plots of regret for OMD-AFW variants over 20 runs for first loss setting (left) and second loss setting (right) for computations in section H.