# OpenReview forum: "Reusing Combinatorial Structure: Faster Iterative Projections over Submodular Base Polytopes"
_NeurIPS.cc/2021/Conference — NeurIPS 2021 Poster_

### Official Review · Reviewer_j61r · 2021-06-25

**Rating:** 7
**Confidence:** 3

**Summary:**

This paper studies projections over submodular base polytopes, which appear as a subroutine of many iterative optimization algorithms. For efficient projections, the authors propose to reuse previous minimizers computed in outer iterations and to exploit combinatorial structures of submodular base polytopes. Some tools for realizing the idea are presented, and those are incorporated into Away-steps Frank--Wolfe to obtain a fast projection method, named Adaptive Away-steps Frank--Wolfe (A2FW). A convergence guarantee and experiments validate the effectiveness of A2FW. A faster projection method based on isotonic optimization is also presented for a special case of cardinality-based submodular polytopes.

--- After rebuttal ---
I appreciate the authors' kind response. After the discussion, I agreed with other reviewers that the experiments are not convincing enough and the contribution is limited. Therefore, I decreased my score to 7, though the technical contribution seems great and I think the paper can be accepted.

**Limitations And Societal Impact:**

The limitations are adequately discussed. The paper seems to have no negative societal impacts.

**Main Review:**

Computing projections over submodular base polytopes is a fundamental subroutine of many iterative optimization methods. To my knowledge, and as mentioned in the paper, the idea of utilizing combinatorial properties of previous projections has not been well studied. I think this paper is great since it addresses an important research subject, introduces an interesting idea, and presents solid theoretical results together with a demonstration of practical utilities.

Below is a list of my comments and questions:

1. In Theorem 6, an advantage over the original AFW seems to be in the pyramidal width, and it is said that this can strictly improve convergence in the case of the unit cube. Is it possible to obtain such lower bounds on the pyramidal width for other common submodular base polytopes?

2. In l. 301, the strong convexity constant, $\mu$, is used when finding tight faces. Since $\mu$ is defined so that the condition of the $\mu$-strong convexity holds globally over $\mathcal{D}$, I guess that the condition, $\tilde{c}_{j+1} - \tilde{c}_j \ge 4L\sqrt{g_t^{\rm FW}} / \mu$, in l. 301 is somewhat pessimistic in practice. Is it possible (and promising) to replace $\mu$ with some parameter that characterizes local strong convexity?

Minor comments:

- In l. 33, it is said that "online Frank-Wolfe is computationally efficient, but has a suboptimal regret of $O(T^{3/4})$ [23, 8]". However, the regret bound is improved to $O(T^{2/3})$ in [8].

- I could not find the definition of $E$ before l. 103, where $1/(2|E|^2)$ appears. It is better to define $E$ somewhere before l. 103.

- l. 140: $\nabla (y)$ -> $\nabla \phi(y)$

- l. 243: doing restricting -> restricting

- l. 276: we instead we just -> instead we just

- l. 406: lipschitz -> Lipschitz

- l. 413: polyak-lojasiewicz -> Polyak-Łojasiewicz

- ll. 446 and 465: wolfe -> Wolfe

- l. 487: frank-wolfe -> Frank-Wolfe

**Time Spent Reviewing:**

3

---

> ### Author Response · Authors · 2021-08-10
> **Response to reviewer j61r**
>
> We thank the reviewer for their detailed reviews and valuable comments, and appreciate the positive feedback on our work. We will carefully implement all writing improvements. We address specific comments next.
>
> Q1. **"Is it possible to obtain such lower bounds on the pyramidal width for other common submodular base polytopes?"**
>
> Answer. We thank the reviewer for this interesting question. This is largely open for interesting combinatorial polytopes, except for simple cases such as the simplex, where Pena and Rodriguez showed in 2019 [1] that the pyramidal width for the simplex restricted to a face $F$ with is $2/\sqrt{\mathrm{dim}(F)}$ (assuming $\mathrm{dim}(F)$ is even for simplicity). For 0-1 submodular polytopes, we suspect that a similar increase in the pyramidal width holds when restricted to a lower dimension face. However, this remains an open question for general submodular polytopes. Thank you for this question, we will add a discussion on this.
>
> Q2. **"In l. 301, the strong convexity constant $\mu$ is used when finding tight faces. Since $\mu$ is defined so that the condition of the $\mu$-strong convexity holds globally over $\mathcal{D}$, I guess that the condition, if $\tilde{c}_{j + 1} - \tilde{c}_j > 4L\sqrt{g_t^{\text{FW}}}/\mu$, in l. 301 is somewhat pessimistic in practice. Is it possible (and promising) to replace  with some parameter that characterizes local strong convexity?"**
>
> Answer. We thank the reviewer for this second interesting open question. We require strong convexity only over the polytope $P$ (and not the domain $\mathcal{D}$). To your point, we do believe it might be possible to improve performance using a local strong convexity estimator. For example, suppose one could construct a series of feasible regions $P \supset P_1 \supset P_2 ... \supset P_k$, such that $P_k$ contains the optimum, and strong convexity improves with $P_i$ as $i$ increases (for example, $h(x) = 1/x$, when optimum is close to 0 in a single dimension). It might be possible to adaptively change the convexity constants, and get further improvements using our toolkits. We will also add a discussion on the quadratic growth condition [2,3] that would also be sufficient to find tight sets and might help relax strong convexity.
>
> **References:**
>
> [1.] Pena, Javier, and Daniel Rodriguez. "Polytope conditioning and linear convergence of the Frank–Wolfe algorithm." Mathematics of Operations Research 44, no. 1 (2019): 1-18.
>
> [2.] Garber, D., Hazan, E. (2015). Faster rates for the frank-wolfe method over strongly-convex sets. In International Conference on Machine Learning (pp. 541-549). PMLR.
>
> [3.] Karimi, H., Nutini, J., Schmidt, M. (2016). Linear convergence of gradient and proximal-gradient methods under the polyak-łojasiewicz condition. In Joint European Conference on Machine Learning and Knowledge Discovery in Databases (pp. 795-811). Springer, Cham.

---

### Official Review · Reviewer_HxTZ · 2021-06-28

**Rating:** 7
**Confidence:** 2

**Summary:**

This paper investigates Frank-Wolfe and Mirror Descent generalizations for combinatorial problems where projections and linear oracles can be efficiently computed or estimated using combinatorial methods. In particular, they look at enhancements when 1) limiting to the active set, which can greatly reduce search space per iteration, 2) relaxations of the combinatorial set, which combined with the first can give convergence if the new iterates are the same.

**Ethical Concerns:**

The paper raises no ethical concerns.

**Limitations And Societal Impact:**

The paper has no negative societal impacts.

**Main Review:**

The problem is interesting and the contribution seems interesting and novel, and very well written. However I am not an expert in this area, and cannot easily catch fundamental issues or closely related existing work.

Originality: Some of these operations, especially over cardinality based objectives, are well-known or easy / intuitive to derive. However, their mixing with frank wolfe and AFW + convergence rates seems novel, and is very interesting. Additionally, the enhancements of rounding or restricting to active set shows interesting benefits in practice.

Quality: I have carefully looked through Lemma 4 proof and it looks reasonable. Lemma 5 proof does not make sense, but I think there are a lot of typos and it may not be a serious problem. Proof of theorem 2 looks reasonable. I did not check the remaining proofs very carefully.

Clarity: I am not 100% in this area but it was not easy for me to verify everything. I do think the paper benefits by spelling out more blatantly what are the key contributions, especially which rates are new, etc. The duality observation for cardinality --> isotonic was also interesting, but it's not clear what the dual can be used for. (That's not really a serious issue though, since duality gaps in FW are usually pretty useful to have together.) An easy example that would make several results more intuitive is to just put in the 0-norm as f and show the calculations there.

Significance: If everything is correct and novel, I think it is a significant enough contribution as the questions posed and strategies given seem very interesting and not that heavily studied. Moreover, submodular optimization has many applications in machine learning.

**Time Spent Reviewing:**

2

---

> ### Author Response · Authors · 2021-08-10
> **Response to reviewer HxTZ**
>
> We thank the reviewer for their  positive review and stating that this work is seems very interesting, not heavily studied and novel. We will carefully revise the writing to address all the issues raised by the reviewer and add an example for computing Euclidean projections over the simplex, with the cardinality set function $f(S)= |S|$ in the appendix (Appendix C.5). We address the main points raised by the reviewer below.
>
> 1. **"I do think the paper benefits by spelling out more blatantly what are the key contributions, especially which rates are new, etc."**
>
> We will highlight our contribution better in the revision: (1) This is the first paper to consider reusing tight inequalities of previous projections to speed up future projections in an iterative optimization setting. (2) We propose a combinatorial toolkit that can exploit structure in previous projections. (3) We explore this in the context of away-step Frank-Wolfe to give a novel A$^2$FW algorithm. Our A$^2$FW enjoys a linear rate of convergence for smooth and strongly convex functions (same as vanilla away-step Frank-Wolfe in the worst-case). (4) A$^2$FW is orders of magnitude faster in practice due to increase in pyramidal width as the iterates converge closer to the optimal solution.
>
> 2. **"What is the use of the dual isotonic optimization problem? (That's not really a serious issue though, since duality gaps in FW are usually pretty useful to have together.)"**
>
> In the paper, we show a duality between Bregman projections and isotonic optimization. We are excited that learning over projections is a dual problem to performing isotonic regression for perturbed data sets, which might be an avenue for future research. We primarily use the dual to compute Bregman projections over cardinality based polytopes efficiently and exactly, by solving the dual problem in $O(n \log n)$ time, and then mapping it back to a primal optimal.
>
> Indeed, as hinted by the reviewer, dual gaps are widely used in iterative optimization methods such as FW variants. The difference between the primal $(P)$ and dual $(D)$ objectives in Theorem 2 gives an upper bound on the primal gap $D_\phi(x,y) - D_\phi(x^*,y)$ for any $x \in B(f)$. This gap might be of interest for a tighter termination criteria than FW gap for computing Bregman projections over cardinality-based polytopes. This is an interesting open question, thank you for pointing this out! We will add this discussion to the paper.
>
> 3. **"I have carefully looked through Lemma 4 proof and it looks reasonable. Lemma 5 proof does not make sense, but I think there are a lot of typos and it may not be a serious problem. Proof of theorem 2 looks reasonable. I did not check the remaining proofs very carefully."**
>
> We thank the reviewer for going over the proofs carefully. We apologize for the variable change error in Lemma 5 (which is proved in [Seuhiro et. al 2012], but stated for completeness in the appendix). This proof should have $y$ instead of $z$ and we will fix that for the camera ready version.
>
> **References:**
>
> 1. D. Suehiro, K. Hatano, S. Kijima, E. Takimoto, and K. Nagano, “Online prediction under submodular constraints,” in International Conference on Algorithmic Learning Theory (ALT). Springer, 2012, pp. 260–274.

---

> > ### Comment · Reviewer_HxTZ · 2021-08-19
> > **response to authors**
> >
> > I have read the authors' response and am satisfied with it. I will keep my score.

---

### Official Review · Reviewer_fArf · 2021-07-16

**Rating:** 6
**Confidence:** 4

**Summary:**

Iterative methods like gradient (and more generally mirror) descent require a projection step which can be expensive. This paper studies how to re-use information from previous projection steps to speed up a projection at the current iteration. In particular, authors focus on the setting where the underlying convex constraint set is a submodular polytope. In this way, the current paper differs from previous work on re-using projection information by working in a combinatorial framework, given by the submodular polytope.

There are two main contributions. First, a $O(n log n)$ time algorithm is given for computing projections over a submodular polytope where the submodular function is so-called cardinality based, i.e. $f(S) = g(|S|)$ for a concave function $g$, which improves upon the best known time of $O(n^2)$. The second contribution is a suite of subroutines for using previous projection information to improve the runtime of current projection. An away-FW variant is proposed which uses these subroutines. Finally, these methods are shown to improve the runtime of mirror descent compared to when vanilla FW is used for the projection step.


**Ethical Concerns:**

All ethical concerns have been addressed and acknowledged.

**Limitations And Societal Impact:**

Yes.

**Main Review:**

The authors clearly explain the research problem (including its significance), the methods used to address the problem, and the extent to which these methods improve current state of the art.
In this sense, the paper is well organized, although there are a number of typos, several of which I list below.

This work is interesting and insightful, as it addresses the broader algorithmic question of moving beyond viewing certain subroutines (here, the projection step) as black boxes. It is unlikely that this “toolkit” provides improvements in worst-case runtime of mirror descent over submodular polytopes, but authors demonstrate how it does indeed improve performance on a simulated example.

There are a few minor downsides to the paper. The first is that while the main contribution seems to be practical algorithmic variants, no open source software is provided. The second downside is that the experimental section is not well-discussed. For example, it is not clear which submodular polytope is being used in the first simulation. The details of the second simulation appear in the appendix and not the main body (although I understand that space constraints may have been an issue here). Finally, I do not understand what the shaded regions in the plots are referring to. Because the main results of the paper are practical algorithmic variants, I think that the paper would benefit if slightly more emphasis was placed on the empirical results.

There are a number of typos or confusing aspects in the paper and I list some here. Authors should review carefully before a camera ready submission.
1. P1 and P2 are often references but never explicitly, mathematically defined. Consider writing them as display equations with tags.
2. Line 64: “and their exist instances” should be “there”
3. Line 75: the current best known runtime can be listed as “O(n^2)” for brevity rather than “O(n log n + n^2)”.
4. Line 101: “The second one rounding tool is algebraic” remove “one”
5. Line 101: “is useful for base polytopes of integral submodular functions”. I believe you should replace “useful for” with “applicable only to”
6. Line 134-138: In definitions of $mu$-strongly convex and $L$-smooth, put the dash outside math mode so that it is short, do not put it inside math mode, i.e. $L-$smooth looks bad but $L$-smooth looks better.
7. Footnote: I believe the definition of strictly convex should have a strict inequality there.
8. Line 189: What does “EO” mean here?
9. Line 216: “We will show this theorem infers a near-optimal number of tight inequalities” what do you mean “near-optimal”? Perhaps use the phrase “most of the tight inequalities” instead.
10. Line 221: “Let $f$ be a monotone submodular with” either add “function” or remove “a”.


**Time Spent Reviewing:**

3

---

> ### Author Response · Authors · 2021-08-10
> **Response for Reviewer fArf**
>
> We thank the reviewer for their detailed feedback, positive comments and careful review our paper, your feedback will help improve our paper immensely. We will write the main class of problems $(P1)$ and $(P2)$ that the paper aims to solve in display equations with tags, and fix the typing errors in the first footnote to have a strict inequality to depict strict convexity. The notation EO (as defined in line 146) is the time for evaluation oracle of the submodular function. We will carefully implement all the suggestions by the reviewer. We address the other major comments raised by the reviewer next:
>
> 1. **"There are a few minor downsides to the paper. The first is that while the main contribution seems to be practical algorithmic variants, no open source software is provided."**
>
> All the code will be made publicly available on GitHub, with a link provided in the camera ready version. We were unable to provide a link to the repository due to anonymity requirements.
>
> 2. **"The second downside is that the experimental section is not well-discussed. For e.g., ..which submodular polytope is being used in the first simulation, ... details of the second simulation in appendix [due to] space constraints.. Finally, I do not understand what the shaded regions in the plots are referring to. Because the main results of the paper are practical algorithmic variants, I think that the paper would benefit if slightly more emphasis was placed on the empirical results."**
>
> We apologize for the short description of the computations. We will expand this discussion significantly, clarify the experimental setup and motivation better, and emphasize the orders of magnitude improvement in running time of OMD, given the additional page in the final version of the paper (currently most of the details are in appendix G). We used the permutahedron with Euclidean distance functions in our experiments in the main body of the paper. The shaded regions in the plots represent the confidence intervals obtained for 20 runs of the same experiment. We agree that our work would benefit with more emphasis on the empirical results and we will add further experiments with larger ground set sizes (n=50,100), with discussion on the trade-offs between the tools.

---

> > ### Comment · Reviewer_fArf · 2021-08-18
> > **reply to comments**
> >
> > Thank you for your thorough reply.
> >
> > One question for you (when revising the paper) is this: what do you mean by "confidence intervals"? This should be expanded upon - perhaps you are not plotting confidence intervals, but something different, e.g. a standard deviation. Either way, this is a minor point that I wanted to bring to your attention.
> >
> > Thanks

---

> > > ### Author Response · Authors · 2021-08-23
> > > **Thanks for catching that!**
> > >
> > > Yes, it is not confidence intervals, rather the deviation in the multiple runs.

---

### Official Review · Reviewer_AdYT · 2021-07-20

**Rating:** 5
**Confidence:** 3

**Summary:**

The paper considers the problem of computing a Bregman projection onto the base polytope of a submodular function. The paper makes two main contributions. The first contribution is to extend existing work for the permutahedron to the base polytope of any cardinality-based submodular function, i.e., a function of the form f(S) = g(|S|) where g is a concave function. Compared to prior work, this results is faster by a factor of n in the number of function evaluations. The second contribution is to design a faster implementation of the Frank-Wolfe algorithm with away steps. This is achieved via a sequence of step optimizations that are designed to reuse information gained in previous iterations in order to detect tight inequalities, reuse active sets, use tight inequalities in the linear optimization step, and round approximate projections to exact projections. The paper evaluates the effectiveness of the proposed speedups on very small scales instances (n=25) where one uses the Bregman projection algorithm as part of an online learning algorithm with the feasible domain being the permutahedron.

**Limitations And Societal Impact:**

Yes

**Main Review:**

Significance: Computing projections onto the submodular base polytope is a challenging problem in practice, and existing algorithms do not scale to large problem instances. The current paper introduces a series of tools that aim to use the combinatorial structure together with the Frank-Wolfe algorithm and obtain improved running times in practice. The correctness of the proposed step optimizations is established via principled proofs. On the negative side, the empirical evaluation is conducted on very small instances (n = 25) and only Frank-Wolfe variants are considered in the experimental evaluation. Thus it remains unclear whether the proposed algorithms scale to larger instances and other choices of base polytopes.

Novelty/originality: The main algorithmic novelty is in the design of the step optimizations used to speed up the Frank-Wolfe algorithm. The result for cardinality-based functions seems to be an extension of prior work for the permutahedron. On a technical level, the proof of correctness for the step optimizations seem to be relatively straightforward. Although the optimizations are a valuable and solid contribution, the paper seems to have somewhat limited algorithmic innovations and the experimental evaluation is quite limited as well.

**Time Spent Reviewing:**

I did not track the hours

---

> ### Author Response · Authors · 2021-08-10
> **Response for Reviewer AdYT**
>
> We thank the reviewer for their review and for also agreeing with our motivation that ``computing projections onto the submodular base polytope is a challenging problem in practice and existing algorithms do not scale to large problem instances''. We address next the main concerns raised about algorithmic novelty and experimental evaluation.
>
> 1. **"The empirical evaluation is conducted on very small instances (n = 25) and only Frank-Wolfe variants are considered in the experimental evaluation. Thus it remains unclear whether the proposed algorithms scale to larger instances and other choices of base polytopes."**
>
> Our primary goal in this paper was to demonstrate how combinatorial properties can help speed-up iterative continuous optimization algorithms, and our experiments were aimed at giving a first evidence that our techniques lead to orders of magnitude speedup. We first note that we did include some experimental results for non-cardinality-based polytopes in Appendix H. We use randomly generated bipartite graphs, and consider the neighborhood function as the general submodular function (e.g., these functions are also used in [Jegelka et. al, NeurIPS 2011]) and show qualitatively similar results.
>
> On the issue of size of the ground sets: our first experiment was in fact performed for $n = 100$. It was aimed at demonstrating that a large fraction of cuts can be inferred by our tool INFER1 once enough points in a neighborhood have been projected. Each of the 500 runs of the experiment had 500 points projected onto the permutahedron with $n=100$.
>
> We acknowledge the reviewer's concern that more evaluation with larger dimensions for the second experiment is needed. We have performed subsequent experiments for $n = 50$ and $n = 100$, we have observed results very similar to those for $n = 25$ reported in the paper with large speed-ups. We will include these experiments in the camera-ready version of the paper, and discuss trade-offs with regret and running time (dependent on the desired accuracy for each projection) in the learning framework.
>
> 2. **"The main algorithmic novelty is in the design of the step optimizations used to speed up the Frank-Wolfe algorithm. The result for cardinality-based functions seems to be an extension of prior work for the permutahedron. Although the optimizations are a valuable and solid contribution, the paper seems to have somewhat limited algorithmic innovations."**
>
> We would first like to emphasize that the key idea of our work is developing tools to reuse combinatorial information from past convex optimizations to speed up future ``near-by'' optimizations, in _any iterative optimization method_ that requires projections, such as Projected Newton's Method, Accerlerated Proximal Gradient, FISTA, and mirorr descent variants.
>
> To compute a projection over the challenging submodular base polytope (with exponential constraints typically), there are mainly two potential ways of doing so: (i) using Frank-Wolfe variants due to attractiveness of linear subproblems, (ii) using combinatorial algorithms such as [Groenevelt, 1991] and [Nagano and Aihara, 2012] (which typically rely on submodular function minimization for detecting tight sets). Our tools for detecting tight sets can be used to speed-up both of these approaches.
>
> The Frank-Wolfe literature has attempted to incorporate the geometry of the problem in various ways, however none of the existing work utilizes combinatorial properties of previous minimizers or detect tight sets with provable guarantees and round to those. To the best of our knowledge, we are the first to adapt AFW to consider improvements over the basic AFW algorithm (Theorem 6). Further, our A$^2$FW allows to combine these tools that reuse structure from past projections (e.g. **T1** and **T3**) and obtain even further computational improvements over the basic AFW algorithm. In our experiments, we report that OMD with our toolkit achieves orders of magnitude improvement in running time, which is significant given that this method can obtain 3-10 times lower regret than Online Frank Wolfe!
>
> Lastly, although a simple extension, we are the first to explicitly state the relation between Bregman projections on _general_ cardinality-based submodular polytopes and isotonic optimization. Lim and Wright [Lim and Wright, 2016] showed that the PAV algorithm could be used to compute projections onto the permutahedron. We extend their result by exploiting submodularity and restriction of minimizers to the optimal face (Lemma 4).
>
> **References:**
>
>     1. Groenevelt, Henri. "Two algorithms for maximizing a separable concave function over a polymatroid feasible region." European Journal of Operations Research 54.2 (1991): 227-236.
>
>     2. Jegelka, Stefanie, Hui Lin, and Jeff A. Bilmes. "On fast approximate submodular minimization." Advances in Neural Information Processing Systems 24 (2011): 460-468.
>
>     3. Lim, Cong Han, and Stephen J. Wright. "Efficient bregman projections onto the permutahedron and related polytopes." Artificial Intelligence and Statistics. PMLR, 2016.
>
>     4. Nagano, Kiyohito, and Kazuyuki Aihara. "Equivalence of convex minimization problems over base polytopes." Japan journal of industrial and applied mathematics 29.3 (2012): 519-534.

---

### Decision · Program_Chairs · 2021-09-28

**Decision:**

Accept (Poster)

**Comment:**

The majority of reviewers thought that the contributions of this paper in terms of efficient projection onto important submodular polytopes and combinatorially-inspired tool-kit for optimization over an important class of such polytopes, which is demonstrated via the introduction of a new Frank-Wolfe method with a novel convergence rate, are interesting and novel beyond the acceptance bar, and may arouse further research along these lines.
I encourage the authors the take the reviewers comments into consideration in the revised version.

**Consistency Experiment:**

NeurIPS has a long history of experimentation. In 2014, NeurIPS ran an experiment in which 10% of submissions were reviewed by two independent committees to quantify the randomness in the review process. This year, we repeated a variant of this experiment to see how the quality of the review process has changed over time.  This paper was part of the experiment and was therefore assigned to two committees (consisting of reviewers, an Area Chair, and a Senior Area Chair) that reached independent decisions.  If both committees made the same recommendation, this recommendation was followed. If a single committee recommended acceptance, the paper was accepted (with the exception of a few cases in which the other committee identified what we considered a fatal flaw, e.g., an error in a key result).

This copy’s committee reached the following decision: **Accept (Poster)**

The other committee assigned to the paper recommended **Reject**.  You can find the other set of reviews, along with any follow up discussion with the authors here:
https://openreview.net/forum?id=OWwm6hzMDsU